# Learned Index with Dynamic $\epsilon$

**Daoyuan Chen**[1,*] **Wuchao Li**[2,*] **Yaliang Li**[1,†] **Bolin Ding**[1] **Kai Zeng**[3,‡] **Defu Lian**[2,†] **Jingren Zhou**[1]
[1] Alibaba Group,   [2] University of Science and Technology of China,   [3] Huawei
{daoyuanchen.cdy, yaliang.li, bolin.ding, jingren.zhou}@alibaba-inc.com
liwuchao@mail.ustc.edu.cn, kai.zeng@huawei.com, liandefu@ustc.edu.cn

## Abstract

Index structure is a fundamental component in database and facilitates broad data retrieval applications. Recent learned index methods show superior performance by learning hidden yet useful data distribution with the help of machine learning, and provide a guarantee that the prediction error is no more than a pre-defined $\epsilon$. However, existing learned index methods adopt a fixed $\epsilon$ for all the learned segments, neglecting the diverse characteristics of different data localities. In this paper, we propose a mathematically-grounded learned index framework with dynamic $\epsilon$, which is efficient and pluggable to existing learned index methods. We theoretically analyze prediction error bounds that link $\epsilon$ with data characteristics for an illustrative learned index method. Under the guidance of the derived bounds, we learn how to vary $\epsilon$ and improve the index performance with a better space-time trade-off. Experiments with real-world datasets and several state-of-the-art methods demonstrate the efficiency, effectiveness and usability of the proposed framework.

## 1 Introduction

Data indexing (Graefe & Kuno, 2011; Wang et al., 2018; Luo & Carey, 2020; Zhou et al., 2020), which stores keys and corresponding payloads with designed structures, supports efficient query operations over data and benefits various data retrieval applications. Recently, Machine Learning (ML) models have been incorporated into the design of index structure, leading to substantial improvements in terms of both storage space and querying efficiency (Kipf et al., 2019; Ferragina & Vinciguerra, 2020a; Vaidya et al., 2021). The key insight behind this trending topic of "learned index" is that the data to be indexed contain useful distribution information and such information can be utilized by trainable ML models that map the keys $\{x\}$ to their stored positions $\{y\}$. To approximate the data distribution, state-of-the-art (SOTA) learned index methods (Galakatos et al., 2019; Kipf et al., 2020; Ferragina & Vinciguerra, 2020b; Stoian et al., 2021) propose to learn piece-wise linear segments $\mathbf{S} = [S_1, ..., S_i, ..., S_N]$, where $S_i : y = a_i x + b_i$ is the linear segment parameterized by $(a_i, b_i)$ and $N$ is the total number of learned segments. These methods introduce an important pre-defined parameter $\epsilon \in \mathbb{Z}_{>1}$ to guarantee the worst-case preciseness: $|S_i(x) - y| \leq \epsilon$ for $i \in [N]$.

By tuning $\epsilon$, various space-time preferences from users can be met. For example, a relatively large $\epsilon$ can result in a small index size while having large prediction errors, and on the other hand, a relatively small $\epsilon$ provides users with small prediction errors while having more learned segments and thus a large index size. Existing learned index methods implicitly assume that the whole dataset to be indexed contains the same characteristics for different localities and thus adopt the same $\epsilon$ for all the learned segments. However, the scenario where there is varied local data distribution, is very common, leading to sub-optimal index performance of existing methods. For example, the real-world *Weblog* dataset used in our experiment has typically non-linear temporal patterns caused by online campus transactions such as class schedule arrangements, weekends and holidays. More importantly, the impact of $\epsilon$ on index performance is intrinsically linked to data characteristics, which are not fully explored and utilized by existing learned index methods.

Motivated by these, in this paper, we theoretically analyze the impact of $\epsilon$ on index performance, and link the characteristics of data localities with the dynamic adjustments of $\epsilon$. Based on the

---

[*]The first two authors contributed equally to this work.
[†]Corresponding author.
[‡]Work was done at Alibaba.

derived theoretical results, we propose an efficient and pluggable learned index framework that dynamically adjusts $\epsilon$ in a principled way. To be specific, under the setting of an illustrative learned index method MET (Ferragina et al., 2020), we present novel analyses about the prediction error bounds of each segment that link $\epsilon$ with the mean and variance of data localities. The segment-wise prediction error embeds the space-time trade-off as it is the product of the *number of covered keys* and *mean absolute error*, which determine the index size and preciseness respectively. The derived mathematical relationships enable our framework to fully explore diverse data localities with an $\epsilon$-learner module, which learns to predict the impact of $\epsilon$ on the index performance and adaptively choose a suitable $\epsilon$ to achieve a better space-time trade-off.

We apply the proposed framework to several SOTA learned index methods, and conduct a series of experiments on three widely adopted real-world datasets. Compared with the original learned index methods with fixed $\epsilon$, our dynamic $\epsilon$ versions achieve significant index performance improvements with better space-time trade-offs. We also conduct various experiments to verify the necessity and effectiveness of the proposed framework, and provide both ablation study and case study to understand how the proposed framework works. Our contributions can be summarized as follows:

- We make the first step to exploit the potential of dynamically adjusting $\epsilon$ for learned indexes, and propose an efficient and pluggable framework that can be applied to a broad class of piece-wise approximation algorithms.
- We provide theoretical analysis for a proxy task modeling the index space-time trade-off, which establishes our $\epsilon$-learner based on the data characteristics and the derived bounds.
- We achieve significant index performance improvements over several SOTA learned index methods on real-world datasets. To facilitate further studies, we make our codes and datasets public at https://github.com/yxdyc/Learned-Index-Dynamic-Epsilon.

## 2 BACKGROUND

**Learned Index.** Given a dataset $\mathcal{D} = \{(x, y) | x \in \mathcal{X}, y \in \mathcal{Y}\}$, $\mathcal{X}$ is the set of *keys* over a universe $\mathcal{U}$ such as reals or integers, and $\mathcal{Y}$ is the set of *positions* where the keys and corresponding payloads are stored. The index such as B$^+$-tree (Abel, 1984) aims to build a compact structure to support efficient query operations over $\mathcal{D}$. Typically, the keys are assumed to be sorted in ascending order to satisfy the *key-position monotonicity*, *i.e.*, for any two keys, $x_i > x_j$ iff their positions $y_i > y_j$, such that the range query ($\mathcal{X} \cap [x_{low}, x_{high}]$) can be handled.

Recently, learned index methods (Kraska et al., 2018; Li et al., 2019; Tang et al., 2020; Dai et al., 2020; Crotty, 2021) leverage ML models to mine useful distribution information from $\mathcal{D}$, and incorporate such information to boost the index performance. To look up a given key $x$, the learned index first predicts position $\hat{y}$ using the learned models, and subsequently finds the stored true position $y$ based on $\hat{y}$ with a binary search or exponential search. By modeling the data distribution information, learned indexes achieve faster query speed and much smaller storage cost than B$^+$-tree index traditional, with different optimization aspects such as on $\epsilon$-bounded linear approximation (Galakatos et al., 2019; Ferragina & Vinciguerra, 2020b; Kipf et al., 2020; Marcus et al., 2020; Li et al., 2021b) and data-layout (Ding et al., 2020; Wu et al., 2021; Zhang & Gao, 2022; Wu et al., 2022; Li et al., 2021a).

$\epsilon$-**bounded Linear Approximation.** Many existing learned index methods adopt piece-wise linear segments to approximate the distribution of $\mathcal{D}$ due to their effectiveness and low computing cost, and introduce the parameter $\epsilon$ to provide a worst-case preciseness guarantee and a tunable knob to meet various space-time trade-off preferences. Here we briefly introduce the SOTA $\epsilon$-bounded learned index methods that are most closely to our work, and refer readers to the literature (Ferragina & Vinciguerra, 2020a; Marcus et al., 2020; Stoian et al., 2021) for details of other methods. We first describe an illustrative learned index algorithm MET (Ferragina et al., 2020). Specifically, for any two consecutive keys of $\mathcal{D}$, suppose their key interval $(x_i - x_{i-1})$ is drawn according to a random process $\{G_i\}_{i \in \mathbb{N}}$, where $G_i$ is a positive independent and identically distributed (i.i.d.) random variable whose mean is $\mu$ and variance is $\sigma^2$. MET learns linear segments $\{S_i : y = a_i x + b_i\}$ via a simple deterministic strategy: the current segment fixes the slope $a_i = 1/\mu$, goes through the first available data point and thus $b_i$ is determined. Then $S_i$ covers the remaining data points one by one until a data point $(x', y')$ gains the prediction error larger than $\epsilon$. The violation triggers a new linear segment that begins from $(x', y')$ and the process repeats until $\mathcal{D}$ has been traversed.

Other $\epsilon$-bounded learned index methods learn linear segments in a similar manner to MET while having different mechanisms to determine the parameters of $\{S_i\}$ such as FITing-Tree (Galakatos et al., 2019), PGM (Ferragina & Vinciguerra, 2020b) and Radix-Spline (Kipf et al., 2020). However, they both constrain all learned segments with the same $\epsilon$. In this paper, we study how to enhance existing learned index methods from a new perspective: dynamic adjustment of $\epsilon$ accounting for diversity of different data localities, and present new theoretical results about the effect of $\epsilon$. In Appx. A, we provide more detailed description and comparison to existing learned index methods.

## 3  LEARN TO VARY $\epsilon$

### 3.1  PROBLEM FORMULATION AND MOTIVATION

Before introducing the proposed framework, we first formulate the task of learning index from data with $\epsilon$ guarantee, and provide some discussions about why we need to vary $\epsilon$. Given a dataset $\mathcal{D}$ to be indexed and an $\epsilon$-bounded learned index algorithm $\mathcal{A}$, we aim to learn linear segments $\mathbf{S} = [S_1, ..., S_i..., S_N]$ with segment-wise varied $[\epsilon_i]_{i \in [N]}$, such that a better trade-off between storage cost (size in KB) and query efficiency (time in ns) can be achieved than the ones using fixed $\epsilon$. Let $\mathcal{D}_i \subset \mathcal{D}$ be the data whose keys are covered by $S_i$, for the remaining data $\mathcal{D} \setminus \bigcup_{j<i} \mathcal{D}_j$, the algorithm $\mathcal{A}$ repeatedly checks whether the prediction error of new data point violates the given $\epsilon_i$ and outputs the learned segment $S_i$. When all the $\epsilon_i$s for $i \in [N]$ take the same value, the problem becomes the one that existing learned index methods are dealing with.

To facilitate theoretical analysis, we focus on two proxy quantities for the target space-time trade-off: (1) the number of learned segments $N$ and (2) the mean absolute prediction error $MAE(\mathcal{D}_i|S_i)$, which is affected and upper-bounded by $\epsilon_i$. We note that the improvements of $N$-$MAE$ trade-off fairly and adequately reflect the improvements of the space-time trade-off: (1) The learned segments size in bytes and $N$ are positively correlated and only different by a constant factor, *e.g.*, the size of a segment can be 128bit if it consists of two double-precision float parameters (slope and intercept); (2) When using exponential search, the querying complexity is $O(\log(N) + \log(MAE(\mathcal{D}_i|S_i)))$, in which the first term indicates the finding process of the specific segment $S'$ that covers the key $x$ for a queried data point $(x, y)$, and the second term indicates the search range $|\hat{y} - y|$ for true position $y$ based on the estimated one $\hat{y} = S'(x)$. In this paper, we adopt exponential search as search algorithm since it is better than binary search for *exploiting the predictive ability of learned models*. In Appx. B, we show that the search range of exponential search is $O(MAE(\mathcal{D}_i|S_i))$, which can be much smaller than the one of binary search, $O(\epsilon_i)$, especially for strong predictive models and the datasets having clear linearity. Similar empirical support can be also found from (Ding et al., 2020).

Now let's examine how the parameter $\epsilon$ affects the $N$-$MAE$ trade-off. We can see that these two performance terms compete with each other and $\epsilon$ plays an important role to balance them. If we adopt a small $\epsilon$, the prediction error constraint is more frequently violated, leading to a large $N$; meanwhile, the preciseness of learned index is improved, leading to a small $MAE$ of the whole data $MAE(\mathcal{D}|\mathbf{S})$. On the other hand, with a large $\epsilon$, we will get a more compact learned index (*i.e.*, a small $N$) with larger prediction errors (*i.e.*, a large $MAE(\mathcal{D}|\mathbf{S})$).

Actually, the effect of $\epsilon$ on index performance is intrinsically linked to the characteristic of the data to be indexed. For real-world datasets, an important observation is that *the linearity degree varies in different data localities*. Recall that we use piece-wise linear segments to fit the data, and $\epsilon$ determines the partition and the fitness of the segments. By varying $\epsilon$, we can adapt to the local variations of $\mathcal{D}$ and adjust the partition such that each learned segment fits the data better. Formally, let's consider the quantity $SegErr_i$ that is defined as the total prediction error within a segment $S_i$, *i.e.*, $SegErr_i \triangleq \sum_{(x,y) \in \mathcal{D}_i} |y - S_i(x)|$, which is also the product of the number of covered keys $Len(\mathcal{D}_i)$ and the mean absolute error $MAE(\mathcal{D}_i|S_i)$. Note that a large $Len(\mathcal{D}_i)$ leads to a small $N$ since $|\mathcal{D}| = \sum_{i=1}^{N} Len(\mathcal{D}_i)$. From this view, the quantity $SegErr_i$ internally reflects the $N$-$MAE$ trade-off. Later we will show how to leverage this quantity to dynamically adjust $\epsilon$.

### 3.2  OVERALL FRAMEWORK

In practice, it is intractable to directly solve the problem formulated in Section 3.1. With a given $\epsilon_i$, the one-pass algorithm $\mathcal{A}$ determines $S_i$ and $\mathcal{D}_i$ until the error bound $\epsilon_i$ is violated. In other words,

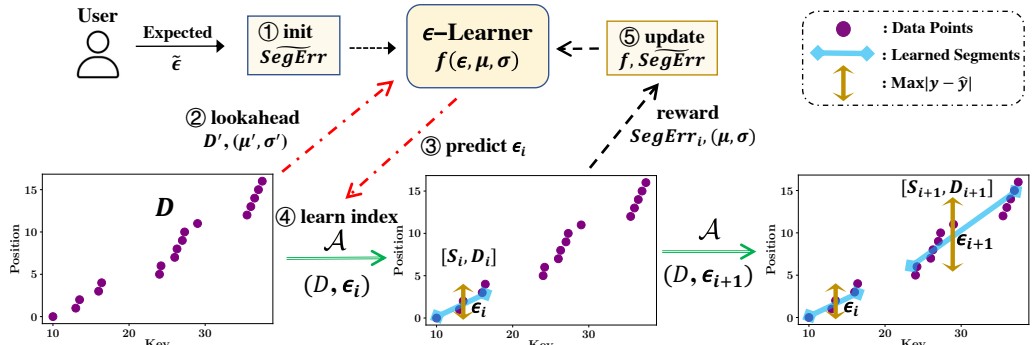

Figure 1: The dynamic $\epsilon$ framework. We ① transform $\tilde{\epsilon}$ into the proxy prediction error $\widetilde{SegErr}$, then ② sample a small look-ahead data $D'$ to estimate the data characteristics $(\mu, \sigma)$. ③ The $\epsilon$-learner predicts a suitable $\epsilon_i$ accordingly, and ④ we learn a new segment $S_i$ using $\mathcal{A}$ (*e.g.*, PGM) with $\epsilon_i$. ⑤ Once $S_i$ triggers the violation of $\epsilon_i$, the $\epsilon$-learner is updated and enhanced with the rewarded ground-truth. Steps ② to ⑤ repeat in an online manner to approximate the distribution of $\mathcal{D}$.

it is unknown what the data partition $\{\mathcal{D}_i\}$ will be *a priori*, which makes it impossible to solve the problem by searching among all the possible $\{\epsilon_i\}$s and learning index with a set of given $\{\epsilon_i\}$.

In this paper, we investigate how to efficiently find an approximate solution to this problem via the introduced $\epsilon$-*learner* module. Instead of heuristically adjusting $\epsilon$, the $\epsilon$-*learner* learns to predict the impact of $\epsilon$ on the index structure and adaptively adjusts $\epsilon$ in a principled way. Meanwhile, the introducing of $\epsilon$-*learner* should not sacrifice the efficiency of the original one-pass learned index algorithms, which is important for real-world practical applications.

These two design considerations establish our dynamic $\epsilon$ framework as shown in Figure 1. The $\epsilon$-*learner* is based on an estimation function $SegErr = f(\epsilon, \mu, \sigma)$ that depicts the mathematical relationships among $\epsilon$, $SegErr_i$ and the characteristics $\mu, \sigma$ of the data to be indexed. As a start, users can provide an expected $\tilde{\epsilon}$ that indicates various preferences under space-sensitive or time-sensitive applications. To meet the user requirements, afterward, we internally transform the $\tilde{\epsilon}$ into another proxy quantity $\widetilde{SegErr}$, which reflects the expected prediction error for each segment if we set $\epsilon_i = \tilde{\epsilon}$. This transformation also links the adjustment of $\epsilon$ and data characteristics together, which enables the data-dependent adjustment of $\epsilon$. Beginning with $\tilde{\epsilon}$, the $\epsilon$-*learner* chooses a suitable $\epsilon_i$ according to current data characteristics, then learns a segment $S_i$ using $\mathcal{A}$, and finally enhances the $\epsilon$-*learner* with the rewarded ground-truth $SegErr_i$ of each segment. To make the introduced adjustment efficient, we propose to only sample a small *Look-ahead* data $\mathcal{D}'$ to estimate the characteristics $(\mu, \sigma)$ of the following data locality. The learning process repeats and is also in an efficient one-pass manner. [1]

Note that the proposed framework provides users the same interface as the ones used by original learned index methods. That is, we do not add any additional cost to the users' experience, and users can smoothly and painlessly use our framework with given $\tilde{\epsilon}$ just as they use the original methods with given $\epsilon$. The $\epsilon$ is an intuitive, meaningful, easy-to-set and method-agnostic quantity for users. On the one hand, we can easily impose restrictions on the worst-case querying cases with $\epsilon$ as the data accessing number in querying process is $O(\log(|\hat{y} - y|))$. On the other hand, $\epsilon$ is easier to estimate than the other quantities such as index size and querying time, which are dependent on specific algorithms, data layouts, implementations and experimental platforms. Our pluggable framework retains the benefits of existing learned index methods, such as the aforementioned usability of $\epsilon$, and the ability to handle dynamic update case and hard size requirement. [2]

We have seen how $\epsilon$ determines index performance and how $SegErr_i$ embeds the *N-MAE* trade-off in Section 3.1. In Section 3.3, we further theoretically analyze the relationship among $\epsilon$, $SegErr_i$, and data characteristics $\mu, \sigma$ at different localities. Based on the analysis, we elaborate on the details of $\epsilon$-*learner* and the internal transformation between $\epsilon$ and $SegErr_i$ in Section 3.4.

---

[1]For better readability, we summarize the notations in Appx. C.

[2]We discuss how to extend existing works in more detail in Appx. D.

## 3.3 Prediction Error Estimation

In this section, we theoretically study the impact of $\epsilon$ on the prediction error $SegErr_i$ of each learned segment $S_i$. The derived closed-form relationships will be taken into account in the design of the proposed $\epsilon$-learner module (Section 3.4). Specifically, for the MET algorithm, we can prove the following theorem to bound the expectation of $SegErr_i$ with $\epsilon$ and the key interval distribution of $\mathcal{D}$.

**Theorem 1.** *Given a dataset $\mathcal{D}$ to be indexed and an $\epsilon$ where $\epsilon \in \mathbb{Z}_{>1}$, consider the setting of the MET algorithm (Ferragina et al., 2020), in which key intervals of $\mathcal{D}$ are drawn from a random process consisting of positive i.i.d. random variables with mean $\mu$ and variance $\sigma^2$, and $\epsilon \gg \sigma/\mu$. For a learned segment $S_i$ and its covered data $\mathcal{D}_i$, denote $SegErr_i = \sum_{(x,y) \in D_i} |y - S_i(x)|$. Then the expectation of $SegErr_i$ satisfies:*

$$\sqrt{\frac{1}{\pi}} \frac{\mu}{\sigma} \epsilon^2 < \mathbb{E}[SegErr_i] < \frac{2}{3} \sqrt{\frac{2}{\pi}} (\frac{5}{3})^{\frac{3}{4}} (\frac{\mu}{\sigma})^2 \epsilon^3.$$

Note that the average length of segments obtained by the MET algorithm is $(\frac{\mu}{\sigma})^2 \epsilon^2$. We now get a constant $\frac{2}{3}\sqrt{\frac{2}{\pi}}(\frac{5}{3})^{\frac{3}{4}} \approx 0.78$ that is tighter than the trivial one with 1 corresponding to the case where each data point reaches the largest error $\epsilon$. This theorem reveals that the prediction error $SegErr_i$ depends on both $\epsilon$ and the data characteristics ($\mu, \sigma$). Recall that $CV = \sigma/\mu$ is the *coefficient of variation*, a classical statistical measure of the relative dispersion of data points. In the context of the linear approximation, the data statistic $1/CV = \mu/\sigma$ in our bounds intrinsically corresponds to the linearity degree of the data. With this, we can find that when $\mu/\sigma$ is large, the data is easy-to-fit with linear segments, and thus we can choose a small $\epsilon$ to achieve precise predictions. On the other hand, when $\mu/\sigma$ is small, it becomes harder to fit the data using a linear segment, and thus $\epsilon$ should be increased to absorb some non-linear data localities. In this way, we can make the total prediction error for different learned segments consistent and achieve a better $N$-*MAE* trade-off. This analysis also confirms the motivation of varying $\epsilon$: The local linearity degrees of the indexed data can be diverse, and we should adjust $\epsilon$ according to the local characteristic of the data, such that the learned index can fit and leverage the data distribution better. In the rest of this section, we provide a proof sketch of this theorem due to the space limitation. For detailed proof, please refer to our Appx. E.

● **Transformed random walk.** The main idea is to model the learning process of linear approximation with $\epsilon$ guarantee as a random walk process, and consider that the absolute prediction error of each data point follows folded normal distributions. Specifically, given a learned segment $S_i : y = a_i x + b_i$, we can calculate the expectation of $SegErr_i$ for this segment as:

$$\mathbb{E}[SegErr_i] = a_i \mathbb{E}\left[\sum_{j=0}^{(j^*-1)} |Z_j|\right] = a_i \sum_{n=1}^{\infty} \mathbb{E}\left[\sum_{j=0}^{n-1} |Z_j|\right] \Pr(j^* = n), \quad (1)$$

where $Z_j$ is the $j$-th position of a transformed random walk $\{Z_j\}_{j \in \mathbb{N}}$, $j^* = \max\{j \in \mathbb{N} | -\epsilon/a_i \leq Z_j \leq \epsilon/a_i\}$ is the random variable indicating the maximal position when the random walk is within the strip of boundary $\pm\epsilon/a_i$, and the last equality is due to the definition of expectation.

● **Proof of upper bound.** Under the MET setting where $a_i = 1/\mu$ and $\epsilon \gg \sigma/\mu$, we find that the increments of the transformed random walk $\{Z_j\}$ have zero mean and variance $\sigma^2$, and many steps are necessary to reach the random walk boundary. With the Central Limit Theorem, we assume the $Z_j$ follows normal distribution with mean $\mu_{zj} = 0$ and variance $\sigma_{zj}^2 = j\sigma^2$, and thus $|Z_j|$ follows the folded normal distribution with expectation $\mathbb{E}(|Z_j|) = \sqrt{2/\pi}\sigma\sqrt{j}$. Thus Eq. (1) becomes:

$$\frac{1}{\mu}\sum_{n=1}^{\infty} \mathbb{E}\left[\sum_{j=0}^{n-1} |Z_j|\right]\Pr(j^*=n) < \frac{1}{\mu}\sum_{n=1}^{\infty}\sum_{j=0}^{n-1}\mathbb{E}[|Z_j|]\Pr(j^*=n) = \frac{\sigma}{\mu}\sqrt{\frac{2}{\pi}}\sum_{n=1}^{\infty}\sum_{j=0}^{n-1}\sqrt{j}\Pr(j^*=n).$$

Using $\mathbb{E}[j^*] = \frac{\mu^2}{\sigma^2}\epsilon^2$ and $Var[j^*] = \frac{2}{3}\frac{\mu^4}{\sigma^4}\epsilon^4$ as derived in Ferragina et al. (2020), we get $\mathbb{E}[(j^*)^2] = \frac{5}{3}\frac{\mu^4}{\sigma^4}\epsilon^4$. With the inequality $\sum_{j=0}^{n-1}\sqrt{j} < \frac{2}{3}n\sqrt{n}$ and $\mathbb{E}[X^{\frac{3}{4}}] \leq (\mathbb{E}[X])^{\frac{3}{4}}$, we get the upper bound:

$$\mathbb{E}[SegErr_i] < \frac{2}{3}\sqrt{\frac{2}{\pi}}\frac{\sigma}{\mu}\mathbb{E}[(j^*)^{\frac{3}{2}}] \leq \frac{2}{3}\sqrt{\frac{2}{\pi}}\frac{\sigma}{\mu}\left(\mathbb{E}[(j^*)^2]\right)^{\frac{3}{4}} = \frac{2}{3}\sqrt{\frac{2}{\pi}}(\frac{5}{3})^{\frac{3}{4}}(\frac{\mu}{\sigma})^2\epsilon^3.$$

• **PROOF OF LOWER BOUND.** Applying the triangle inequality into Eq. (1), we can get $\mathbb{E}[SegErr_i] > \frac{1}{\mu}\sum_{n=1}^{\infty}\mathbb{E}[|Z|]\Pr(j^* = n)$, where $Z = \sum_{j=0}^{n-1}Z_j$, and $Z$ follows the normal distribution since $Z_j \sim N(0, \sigma_{zj}^2)$. We can prove that $|Z|$ follows the folded normal distribution whose expectation $\mathbb{E}[|Z|] > \sigma(n-1)/\sqrt{\pi}$. Thus the lower bound is:

$$\mathbb{E}[SegErr_i] > \frac{\sigma}{\mu}\sqrt{\frac{1}{\pi}}\sum_{n=1}^{\infty}(n-1)\Pr(j^* = n) = \frac{\sigma}{\mu}\sqrt{\frac{1}{\pi}}\mathbb{E}[j^* - 1] = \sqrt{\frac{1}{\pi}}(\frac{\mu}{\sigma}\epsilon^2 - \frac{\sigma}{\mu}).$$

Since $\epsilon \gg \frac{\sigma}{\mu}$, we can omit the right term $\sqrt{1/\pi} \cdot \sigma/\mu$ and finish the proof. Although the derivations are based on the MET algorithm whose slope is the reciprocal of $\mu$, we found that the mathematical forms among $\epsilon$, $\mu/\sigma$ and $SegErr_i$ are still applicable to other $\epsilon$-bounded methods, and further prove that the learned segment slopes of these methods are similar with bounded differences in Appx. F.

### 3.4  $\epsilon$-LEARNER

Now given an $\epsilon$, we have obtained the closed-form bounds of the $SegErr$ in Theorem 1, and both the upper and lower bounds are in the form of $w_1(\frac{\mu}{\sigma})^{w_2}\epsilon^{w_3}$, where $w_{1,2,3}$ are some coefficients. As the concrete values of these coefficients can be different for different datasets and different methods, we propose to learn the following trainable estimator to make the error prediction more precise:

$$SegErr = f(\epsilon, \mu, \sigma) = w_1(\frac{\mu}{\sigma})^{w_2}\epsilon^{w_3},$$

$$s.t. \sqrt{\frac{1}{\pi}} \leq w_1 \leq \frac{2}{3}\sqrt{\frac{2}{\pi}}(\frac{5}{3})^{\frac{3}{4}}, \quad 1 \leq w_2 \leq 2, \quad 2 \leq w_3 \leq 3. \tag{2}$$

With this learnable estimator, we feed data characteristic $\mu/\sigma$ of the look-ahead data and the transformed $\widetilde{SegErr}$ into it and find a suitable $\epsilon^*$ as $\left(\widetilde{SegErr}/w_1(\frac{\mu}{\sigma})^{w_2}\right)^{1/w_3}$. We will discuss the look-ahead data and the transformed $\widetilde{SegErr}$ in the following paragraphs. Now let's discuss the reasons for how this adjustment can achieve better index performance. Actually, the $\epsilon$-learner proactively plans the allocations of the total prediction error indicated by the user (*i.e.*, $\tilde{\epsilon} \cdot |\mathcal{D}|$) and calculates the tolerated $\widetilde{SegErr}$ for the next segment. By adjusting current $\epsilon$ to $\epsilon^*$, the following learned segment can fully utilize the distribution information of the data and achieve better performance in terms of *N-MAE* trade-off. To be specific, when $\mu/\sigma$ is large, the local data has clear linearity, and thus we can adjust $\epsilon$ to a relatively small value to gain precise predictions; although the number of data points covered by this segment may decrease and then the number of total segments increases, such cost paid in terms of space is not larger than the benefit we gain in terms of precise predictions. Similarly, when $\mu/\sigma$ is small, $\epsilon$ should be adjusted to a relatively large value to lower the learning difficulty and absorb some non-linear data localities; in this case, we gain in terms of space while paying some costs in terms of prediction accuracy. The segment-wise adjustment of $\epsilon$ improves the overall index performance by continually and data-dependently balancing the cost of space and preciseness.

**Look-ahead Data.**  To make the training and inference of the $\epsilon$-learner light-weight, we propose to look ahead a few data $\mathcal{D}'$ to reflect the characteristics of the following data localities. Specifically, we leverage a small subset $\mathcal{D}' \subset \mathcal{D} \setminus \bigcup_{j<i}\mathcal{D}_j$ to estimate the value $\mu/\sigma$ for the following data. In practice, we set the size of $\mathcal{D}'$ to be 404 when learning the first segment as initialization, and $\left(\frac{1}{(i-1)}\sum_{j=1}^{i-1}Len(\mathcal{D}_j)\right) \cdot \rho$ for the other following segments. Here $\rho$ is a pre-defined parameter indicating the percentage that is relative to the average number of covered keys for learned segments, considering that the distribution of $\mu/\sigma$ can be quite different to various datasets. As for the first segment, according to the literature (Kelley, 2007), the sample size 404 can provide a 90% confidence interval for a coefficient of variance $\sigma/\mu \leq 0.2$.

$\widetilde{SegErr}$ **and Optimization.**  As aforementioned, taking the user-expected $\tilde{\epsilon}$ as input, we aim to reflect the impact of $\tilde{\epsilon}$ with a transformed proxy quantity $\widetilde{SegErr}$ such that the $\epsilon$-learner can choose suitable $\epsilon^*$ to meet users' preference while achieving better *N-MAE* trade-off. Specifically, we make the value of $\widetilde{SegErr}$ updatable, and update it to be $\widetilde{SegErr} = w_1(\hat{\mu}/\hat{\sigma})^{w_2}\tilde{\epsilon}^{w_3}$ once a new segment is learned, where $\hat{\mu}/\hat{\sigma}$ is the mean value of all the processed data so far. This strategy enables us to promptly incorporate both the user preference and the data distribution into the calculation of

$\widetilde{SegErr}$. As for the optimization of the light-weight model, *i.e.*, $f(\epsilon, \mu, \sigma)$ that contains only three learnable parameters $w_{1,2,3}$, we adopt the projected gradient descent den Hertog & Roos (1991) with the parameter constraints in Eq. (2). In this way, we only need to track a few statistics and learn the $\epsilon$ estimator in an efficient one-pass manner. The overall algorithm is summarized in Appx. G.

## 4 EXPERIMENTS

### 4.1 EXPERIMENTAL SETTINGS

**Baselines and Metrics.** We apply our framework to several SOTA learned index methods, including *MET* (Ferragina et al., 2020), *FITing-Tree* (Galakatos et al., 2019), *Radix-Spline*, and *PGM* (Ferragina & Vinciguerra, 2020b). For evaluation, we consider the index performance in terms of its learned segments $N$, size in bytes, prediction preciseness *MAE*, and the total querying time in ns. For a quantitative comparison w.r.t. the trade-off improvements, we calculate the **A**rea **U**nder the $N$-$MAE$ **C**urve (AUNEC) where the x-axis and y-axis indicate $N$ and *MAE* respectively. For AUNEC metric, the smaller, the better. More introduction and implementation details are in Appx. H.

**Datasets.** We use several widely adopted datasets that are from real-world applications and differ in data scales and distributions (Kraska et al., 2018; Galakatos et al., 2019; Ding et al., 2020; Ferragina & Vinciguerra, 2020b; Li et al., 2021b), including *Weblogs* and *IoT* (timestamp keys), *Map* (location coordinate keys), and *Lognormal* (synthetic keys). More details and visualization are in Appx. I.

### 4.2 OVERALL INDEX PERFORMANCE

$N$-*MAE* **Trade-off Improvements.** In Table 1, we summarize AUNEC improvements in percentage brought by the proposed framework of all the baseline methods on all the datasets. We also illustrate the $N$-*MAE* trade-off curves for some cases in Figure 2, where the blue curves indicate results achieved by fixed $\epsilon$ version while the red curves are for dynamic $\epsilon$. Other baselines and datasets yield similar curves, which we include in Appx. J.1 due to the space limitation. These results show that the dynamic $\epsilon$ versions of all the baseline methods achieve much better $N$-*MAE* trade-off ($-15.66\%$ to $-22.61\%$ averaged improvements as smaller AUNEC indicates better performance), demonstrating the effectiveness and wide applicability of the proposed framework. As discussed in previous sections, datasets usually have diverse key distributions at different data localities, and the proposed framework can data-dependently adjust $\epsilon$ to fully utilize the distribution information of data localities and thus achieve better index performance in terms of $N$-*MAE* trade-off. Here the Map dataset has significant non-linearity caused by spatial characteristics, and it is hard to fit using linear segments (all baseline methods learn linear segments), thus relatively small improvements are achieved.

Table 1: The AUNEC relative *improvements* for learned index methods with dynamic $\epsilon$.

| | Weblogs | IoT | Map | Lognormal | Average |
|---|---|---|---|---|---|
| MET | -25.87% | -7.66% | -7.63% | -21.48% | -15.66% |
| FITing-Tree | -31.18% | -25.56% | -4.94% | -28.24% | -22.48% |
| Radix-Spline | -28.37% | -24.59% | -6.14% | -31.32% | -22.61% |
| PGM | -22.42% | -25.01% | -7.18% | -6.52% | -15.28% |

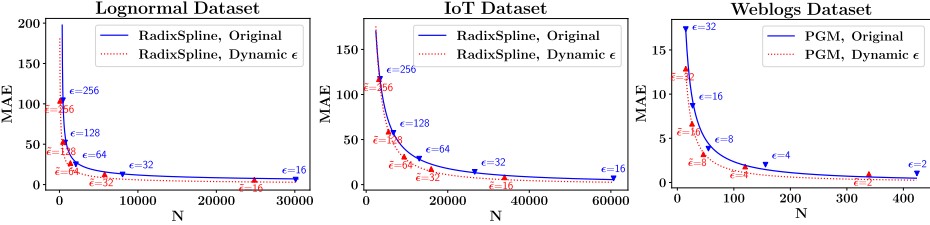

Figure 2: The $N$-*MAE* trade-off curves for learned index methods.

**Querying Time Improvements.** Recall that the querying time of each data point is in $O(\log(N) + \log(|y - \hat{y}|)$ as we mentioned in Section 3.1, where $N$ and $|y - \hat{y}|$ are inversely impacted by $\epsilon$. To

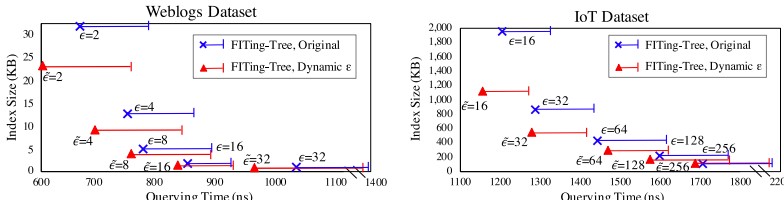

Figure 3: Improvements in terms of querying times for learned index methods with dynamic $\epsilon$.

examine whether the performance improvements w.r.t. $N$-*MAE* trade-off (*i.e.*, Table 1) can lead to better querying efficiency in real-world systems, we show the averaged total querying time per query and the actual learned index size in bytes for two scenarios in Figure 3. We also mark the 99th percentile (P99) latency as the right bar. We can observe that the dynamic $\epsilon$ versions indeed gain faster average querying speed, since we improve both the term $N$ as well as the term $|y - \hat{y}|$ via adaptive adjustment of $\epsilon$. Besides, we find that the dynamic version achieves comparable or even better P99 results than the static version, due to the fact that our method effectively adjusts $\epsilon$ based on the expected $\tilde{\epsilon}$ and data characteristic, making the $\{\epsilon_i\}$ fluctuated within a moderate range and leading to good robustness. A similar conclusion can be drawn from other baselines and datasets, and we present their results in Appx. J.1. Another thing to note is that, this experiment also verifies the usability of our framework in which users can flexibly set the expected $\tilde{\epsilon}$ to meet various space-time preferences just as they set $\epsilon$ in the original learned index methods.

**Index Building Cost.** Compared with the original learned index methods that adopt a fixed $\epsilon$, we introduce extra computation to dynamically adjust $\epsilon$ in the index building stage. Does this affect the efficiency of the original methods? Here we report the relative increments of building times in Table 2. From it, we can observe that the proposed dynamic $\epsilon$ framework achieves comparable building times to all the original learned index methods on all the datasets, showing the efficiency of our framework since it retains the online learning manner with the same complexity as the original methods (both in $O(|\mathcal{D}|)$). Note that we only need to pay this extra cost once, *i.e.*, building the index once, and then the index structures can accelerate the frequent data querying operations for real-world applications.

Table 2: Building time *increments* in percentage for learned index methods with dynamic $\epsilon$.

|  | Weblogs | IoT | Map | Lognormal | Average |
|---|---|---|---|---|---|
| MET | 10.54% | 5.14% | 8.33% | 5.26% | 7.32% |
| FITing-Tree | 10.70% | 1.88% | 5.35% | 5.23% | 5.79% |
| Radix-Spline | 10.19% | 1.64% | 3.85% | 8.96% | 6.16% |
| PGM | 16.76% | 2.20% | 1.28% | 21.29% | 10.38% |

## 4.3 ABLATION STUDY OF DYNAMIC $\epsilon$

To gain further insights into how the proposed dynamic $\epsilon$ framework works, we compare the proposed one with three dynamic $\epsilon$ variants: (1) *Random $\epsilon$* is a vanilla version that randomly choose $\epsilon$ from $[0, 2\tilde{\epsilon}]$ when learning each new segment; (2) *Polynomial Learner* differs our framework with another polynomial function $SegErr(\epsilon) = \theta_1 \epsilon^{\theta_2}$ where $\theta_1$ and $\theta_2$ are trainable parameters; (3) *Least Square Learner* differs our framework with an optimal (but very costly) strategy to learn $f(\epsilon, \mu, \sigma)$ with the least square regression.

Table 3: The AUNEC relative changes of dynamic $\epsilon$ variants compared to the proposed framework.

|  | Weblogs | IoT | Map | Lognormal | Average |
|---|---|---|---|---|---|
| Random $\epsilon$ | +70.94% | +68.19% | +53.29% | +73.38% | +66.45% |
| Polynomial Learner | +49.32% | +40.57% | +7.71% | +42.77% | +35.09% |
| Least Square Learner | +4.44% | +9.32% | +2.04% | −17.63% | −0.46% |

We summarize the AUNEC changes in percentage compared to the proposed framework in Table 3. Here we only report the results for FITing-Tree due to the space limitation and similar results can be observed for other methods. Recall that for AUNEC, the smaller, the better. From this table, we have the following observations: (1) The *Random $\epsilon$* version achieves much worse results than the proposed dynamic $\epsilon$ framework, showing the necessity and effectiveness of learning the impact of $\epsilon$. (2) The *Polynomial Learner* achieves better results than the *Random $\epsilon$* version while still having a

large performance gap compared to our proposed framework. This indicates the usefulness of the derived theoretical results that link the index performance, the $\epsilon$, and the data characteristics together. (3) For the *Least Square Learner*, we can see that it achieves similar AUNEC results compared with the proposed framework. However, it has higher computational complexity and pays the cost of much larger building times, *e.g.*, $14\times$ and $53\times$ longer building times on IoT and Map respectively. These results demonstrate the effectiveness and efficiency of the proposed framework that adjusts $\epsilon$ based on the theoretical results, which will be validated next.

### 4.4 CASE STUDY

We visualize the partial learned segments for FITing-Tree with fixed and dynamic $\epsilon$ on IoT dataset in Figure 4, where the $N$ and $\sum SegErr_i$ indicates the number of learned segments and the total prediction error for the shown segments respectively. The $\overrightarrow{\mu/\sigma}$ indicates the characteristics of covered data $\{\mathcal{D}_i\}$. We can see that our dynamic framework helps the learned index gain both smaller space (7 v.s. 4) and smaller total prediction errors (48017 v.s. 29854). Note that $\epsilon$s within $\overrightarrow{\epsilon_i}$ are diverse due to the diverse linearity of different data localities: For the data whose positions are within about $[30000, 30600]$ and $[34700, 35000]$, the proposed framework chooses large $\epsilon$s as their $\mu/\sigma$s are small, and by doing so, it achieves smaller $N$ than the fixed version by absorbing these non-linear localities; For the data in the middle part, they have clear linearity with large $\mu/\sigma$s, and thus the proposed framework adjusts $\epsilon$ as 19 and 10 that are smaller than 32 to achieve better precision. These experimental observations are consistent with our analysis in the paragraph under Eq. (2),

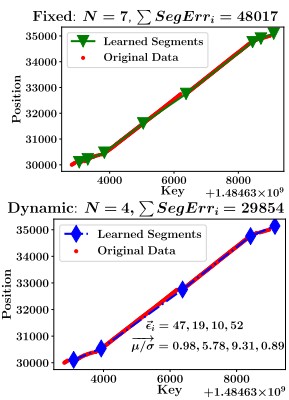

Figure 4: Visualization of the learned index (partial) on IoT for FITing-Tree with fixed $\epsilon = 32$ and dynamic version ($\tilde{\epsilon} = 32$).

and clearly confirm that the proposed framework adaptively adjusts $\epsilon$ based on data characteristics.

### 4.5 MORE EXPERIMENTS IN APPENDIX

Due to the space limitation, we provide further experiments and analysis in Appendix, including:

- More results about the overall index performance (Appx.J.1) and ablation studies (Appx.J.2) on other datasets and methods, which support similar conclusions in Sec.4.2 and Sec.4.4.
- The theoretical validation (Appx.J.3) for Theorem 1 that $SegErr_i$ is within the derived bounds, and Theorem 2 that the learned slopes of various $\epsilon$-bounded methods have the same trends and both concentrate on $1/\mu_i$ with a bounded difference. This shows that baselines have the same mathematical forms as we derived, and the proposed $\epsilon$-learner works well with wide applicability.
- The insights about which kinds of datasets will benefit from our dynamic adjustment (Appx.J.4), with the help of an indicative quantity, the *coefficient of variation* value ($\sigma/\mu$).

### 5 CONCLUSIONS

Existing learned index methods introduce an important hyper-parameter $\epsilon$ to provide a worst-case preciseness guarantee and meet various space-time user preferences. In this paper, we provide formal analyses about the relationships among $\epsilon$, data local characteristics, and the introduced quantity $SegErr_i$ for each learned segment, which is the product of the number of covered keys and *MAE*, and thus embeds the space-time trade-off. Based on the derived bounds, we present a pluggable dynamic $\epsilon$ framework that leverages an $\epsilon$ learner to data-dependently adjust $\epsilon$ and achieve better index performance in terms of space-time trade-off. A series of experiments verify the effectiveness, efficiency, and usability of the proposed framework.

We believe that our work contributes a deeper understanding of how the $\epsilon$ impacts the index performance, and enlightens the exploration of fine-grained trade-off adjustments by considering data local characteristics. Our study also opens several interesting future works. For example, we can apply the proposed framework to other problems in which the piece-wise approximation algorithms with fixed $\epsilon$ are used while still requiring space-time trade-off, such as similarity search and lossy compression for time series data (Chen et al., 2007; Xie et al., 2014; Buragohain et al., 2007; O'Rourke, 1981).

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

APPENDIX FOR THE PAPER: LEARNED INDEX WITH DYNAMIC $\epsilon$

Our appendix includes the following content:

- Sec.A: further **descriptions** and **comparison** about the related $\epsilon$-based learned index methods and data layout optimization-based methods.
- Sec.B: the details of the binary search and exponential search, and the connections between **prediction error** and these specific **searching strategies**.
- Sec.C: the **notations** adopted in this paper.
- Sec.D: the discussion about how the proposed framework **inherits the good abilities** of existing learned index methods.
- Sec.E: the full **proof** of Theorem 1.
- Sec.F: the analysis about the **learned slopes** of other $\epsilon$-bounded methods.
- Sec.G: the summarized **algorithm** of the proposed method.
- Sec.H: the **implementation details** of experiments.
- Sec.I: the detailed descriptions and visualization of the adopted **datasets**.
- Sec.J: more **experimental results** including the overall index performance and ablation study on other datasets and methods (Sec.J.1 and Sec.J.2), and the theoretical validation (Sec.J.3). Besides, we explore an indicative quantity (the $CV$ value) to provide further insight into the rationale of the proposed framework (Sec.J.4).

## A MORE DETAILS ABOUT RELATED WORKS

### A.1 $\epsilon$-BOUNDED LINEAR APPROXIMATION METHODS

Besides the MET method mentioned in Sec.2, we give more introduction to the related $\epsilon$-bounded linear approximation methods. FITing-Tree (Galakatos et al., 2019) uses a greedy shrinking cone algorithm. PGM (Ferragina & Vinciguerra, 2020b) adopts another one-pass algorithm that achieves the optimal number of learned segments. Radix-Spline (Kipf et al., 2020) introduces a radix structure to organize learned segments. Here we use a toy dataset to demonstrate the workflow of the $\epsilon$-bounded linear approximation methods. Suppose we study every segment with a demonstrative method in an online manner, which is simplified from MET and expresses the general idea of the class of the $\epsilon$-bounded linear approximation methods: When a new data point comes over, we connect it with the starting point of the segment as the linear function of this segment, and check whether there is a data point whose prediction error is greater than $\epsilon$. If so, the learning of this segment is terminated, and the current data point will serve as the starting point for a new segment. The first two subfigures in Figure 5 show the learning process of the first segment, and the data point which terminates the process is taken as the starting point of the second segment. Subsequent subfigures show the learning process for the following segments just as the first segment. Existing works are different in determining linear functions and termination conditions, but they all follow a similar flow like this.

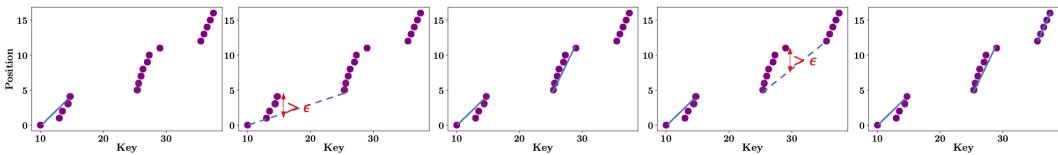

Figure 5: Workflow of $\epsilon$-bounded linear approximation method.

However, existing methods constrain all learned segments with the same $\epsilon$. All of these piece-wise segments based approaches attempt to improve performance by changing the way segments are learned or organized, but ignore the optimization potential of dynamically varying $\epsilon$. In this paper, we discuss the impact of $\epsilon$ in more depth and investigate how to enhance existing learned index methods from a new perspective: dynamic adjustment of $\epsilon$ accounting for the diversity of different

data localities, as shown in our mathematical derivation linking the indexing performance, $\epsilon$ and data statistics $\mu/\sigma$ (Sec. 3.3) and the proposed $\epsilon$-learner (Sec. 3.4). Besides, different from (Ferragina et al., 2020) which reveals the relationship between $\epsilon$ and index size performance based on MET. In Sec.3.3, we give novel analyses about the impact of $\epsilon$ on not only index size, *but also index preciseness and a comprehensive trade-off quantity*, which facilitates the proposed dynamic $\epsilon$ adjustment.

## A.2 DATA-LAYOUT-OPTIMIZATION BASED METHODS

In this paper, we mainly focus on the $\epsilon$-based learned index methods. In recent years, some data-layout-optimization based methods also gain promising indexing performance. For example, ALEX (Ding et al., 2020) improves Recursive Model Index (RMI) Kraska et al. (2018) by reserving gaps within arrays to enhance the ability to update indexing case (insert, delete, update, etc.,). LIPP (Wu et al., 2021) proposes to extend the tree structure with zero prediction error for update operations and proposes an adjustment strategy to provide a bounded height of the tree index. NFL (Wu et al., 2022) proposes to transform the complex key distribution into a near-uniform distribution. CARMI (Zhang & Gao, 2022) leverages an entropy-based cost model to improve the data partitioning of tree nodes in learned indexes. FINEdex (Li et al., 2021a) focuses on concurrent and independent model processing with the help of a fattened data structure. These two performance optimization perspectives are relatively orthogonal. Both types of learned index methods have their pros and cons:

**Worst-case guarantee.** For the $\epsilon$-bounded methods (Ferragina & Vinciguerra, 2020b; Stoian et al., 2021; Galakatos et al., 2019; Ferragina et al., 2020; Kipf et al., 2020), the most important advantage they provide is the worst-case guarantee in each segment. This property is fairly valuable in many realistic indexing applications such as financial databases and on-device intelligence. Also note that our approach still maintains comparable 99th percentile (P99) performance compared to static $\epsilon$ baselines as shown in the overall index performance comparison (Figure 3 and Figure 10).

**Index Size.** Data-layout-based approaches such as ALEX (Ding et al., 2020) do achieve better indexing performance in dynamic scenarios, but it pays a larger index size overhead because of the introduced gap insertion technique (reserving empty space for possibly inserted data). Empirically, we examine the performance of ALEX with our experimental settings and find that it gains comparable query time and larger index size than the learned index with dynamic $\epsilon$ as the following Table 4 shows, where the $q_{time}$ and index $size$ are in ns and KB respectively. We adopt the default hyper-parameters of ALEX and the $\epsilon$ is 4, 32, 64, and 32 in these four datasets respectively for RadixSpline.

We note that the index size is still very important even with a large amount of memory available today, as shown by the fact that almost all related learned index works examine this metric in their experiments. It is worth noting that index size can have a significant impact on latency indirectly, since the smaller the index, the more it can fit into the hierarchical high-speed CPU caches, and thus increase the hit rates to speed up the overall indexing performance Zhang & Gao (2022); Zhang et al. (2021). Moreover, in some practical applications, the available memory can be relatively small such as in cases where users need to build indexes from multiple keys of the data, and need to use the index on IoT devices for edge computation.

**Dynamic Hyper-parameter.** Moreover, our novel framework can be regarded as an automatic method that determines hyper-parameters ($\epsilon$) according to the varied local properties of the data. Although the data layout optimization goes beyond our scope, our insight can also contribute to the ALEX approach, since it also introduces hyper-parameters in local linear segments learning, the lower and upper density limits on each gapped array: $d_l, d_u \in (0, 1]$. The authors empirically set them to be 0.6 and 0.8 respectively. However, different gapped arrays may gain better performance with different density limitations, since the data distribution can be varied across different localities, as we have shown in experiments.

## B CONNECTING PREDICTION ERROR WITH SEARCHING STRATEGY

As we mentioned in Section 3.1, we can find the true position of the queried data point in $O(\log(N) + \log(|\hat{y} - y|))$ where $N$ is the number of learned segments and $|\hat{y} - y|$ is the absolute prediction error. A binary search or exponential search finds the stored true position $y$ based on $\hat{y}$. It is worth pointing

Table 4: The indexing performance comparison between ALEX and RadixSpline with the proposed dynamic $\epsilon$ framework.

| | Weblogs | | IoT | | Map | | Lognormal | |
|---|---|---|---|---|---|---|---|---|
| | $q_{time}$ | $size$ | $q_{time}$ | $size$ | $q_{time}$ | $size$ | $q_{time}$ | $size$ |
| ALEX | 713 | 1428 | 915 | 1632 | 521 | 582 | 962 | 1321 |
| RadixSpline, Dynamic $\epsilon$ | 602 | 995 | 974 | 1275 | 428 | 373 | 937 | 1085 |

out that the searching cost in terms of searching range $|\hat{y} - y|$ of binary search strategy corresponds to the maximum absolute prediction error $\epsilon$, whereas the one of exponential search corresponds to the mean absolute prediction error (*MAE*). In this paper, we decouple the quantity $SegErr_i$ as the product of $Len(\mathcal{D}_i)$ and $MAE(\mathcal{D}_i|S_i)$ in the derivation of Theorem 1. Built upon the theoretical analysis, we adopt exponential search in experiments to better leverage the predictive models.

To clarify, let's consider a learned segment $S_i$ with its covered data $\mathcal{D}_i$. Let $|\hat{y_k} - y_k|$ be the absolute prediction error of $k$-th data point covered by this segment, and $\epsilon_i$ be the maximum absolute prediction error of $S_i$, *i.e.*, $|\hat{y_k} - y_k| \leq \epsilon_i$ for all $k \in [len(\mathcal{D}_i)]$.

- The binary search is conducted within the searching range $[\hat{y_k} \pm \epsilon_i]$ for each data point [3], thus the mean search range is $\frac{1}{len(\mathcal{D}_i)} \sum_{k=1}^{len(\mathcal{D}_i)} 2\epsilon_i = O(\epsilon_i)$, which is independent of the preciseness of the learned segment and an upper bound of $MAE(\mathcal{D}_i|S_i)$.

- The exponential search first finds the searching range where the queried data may exist by centering around the $\hat{y}$, repeatedly doubling the range $[\hat{y} \pm 2^q]$ where the integer $q$ grows from 0, and comparing the queried data with the data points at positions $\hat{y} \pm 2^q$. After finding the specific range such that a $q_k$ satisfies $2^{\log(q_k)-1} \leq |\hat{y_k} - y_k| \leq 2^{\lceil \log(q_k) \rceil}$ for the $k$-th data, a binary search is conducted to find the exact location. In this way, the mean search range is $\frac{1}{len(\mathcal{D}_i)} \sum_{k=1}^{len(\mathcal{D}_i)} (2^{\lceil \log(q_k) \rceil + 1}) = O(MAE(\mathcal{D}_i|S_i))$, which can be much smaller than $O(\epsilon_i)$, especially for strong predictive models and the datasets having clear linearity.

## C NOTATIONS

We summarize the adopted notations in Table 5 for convenience.

Table 5: The adopted Notations

| Notation | Description |
|---|---|
| $N$ | The total number of segments |
| $S_i$ | The i-th segment of learned index |
| $\mathcal{D}$ | The whole data to be indexed |
| $\mathcal{D}_i$ | The data covered by the i-th segment |
| $\tilde{\epsilon}$ | Expected $\epsilon$ given by user |
| $\epsilon_i$ | The maximum prediction error of segment $i$ |
| $SegErr_i$ | The sum of the errors for the data covered by the i-th segment |
| $\widetilde{SegErr}$ | Expected segment error calculated based on $\tilde{\epsilon}$ |
| $\mathcal{D}'$ | Sampled data to reflect the characteristics of the following data localities. |
| $\rho$ | The hyperparameter to determine the size of $\mathcal{D}'$ |
| $\hat{\mu}$ | Mean of data intervals in $\mathcal{D}'$ |
| $\hat{\sigma}$ | Standard deviation of data intervals in $\mathcal{D}'$ |
| $f(\cdot)$ | The learnable function for determine $\epsilon_i$ |
| $w_1, w_2, w_3$ | Learnable parameters of the *epsilon*-learner ($f$) |

---

[3]The lower bound and upper bounds of searching ranges should be constricted to 0 and $len(\mathcal{D}_i)$ respectively. For brevity, we omit the corner cases when comparing these two searching strategies as they both need to handle the out-of-bounds scenario.

# D   INHERITING THE ABILITIES OF EXISTING WORKS

In this Section, we discuss the benefits of our proposed framework brought by its pluggable property with two example scenarios, the dynamic data update and hard limitation on user-required index size.

We note that the data insert operation has been discussed in the adopted baseline methods, FITing-Tree Galakatos et al. (2019) and PGM Ferragina & Vinciguerra (2020b). More importantly, neither of these two methods altered the notion of $\epsilon$ when dealing with the data insertion, and they still relied on their $\epsilon$-bounded piece-wise segmentation algorithms. The proposed framework is still valid when using their respective solutions to handle data insertion. Specifically, FITing-Tree proposes to introduce a buffer for each learned segment, which is used to store the inserted keys, and when the buffer is full, the data covered by the segment will be re-segmented (see Section 5 in Galakatos et al. (2017)). PGM adopts a logarithmic method O'Neil et al. (1996); Overmars (1987) that maintains a series of sorted sets $\{S_0, S_1, ..., S_b\}$ where $b = \theta(\log(|\mathcal{D}|))$, and builds multiple PGM-INDEX models over the sets. When a key $x$ is inserted, a new PGM-INDEX will be built over the merged sets (see Section 3 in Ferragina & Vinciguerra (2020b)). In general, these solutions proposed by existing methods for inserting keys are based on *re-indexing for a piece of data along with the inserted data*, and the *re-indexing processes are the same as the original piece-wise linear segmentation processes* but for different data, therefore, we can still apply the proposed dynamic-$\epsilon$ framework for these methods in insertion scenarios just like we adjust $\epsilon$ and learn index according to the new data to be re-indexed.

For the hard size limitation case, we observe that the existing work PGM introduced a multi-criteria variant that auto-tunes itself with pre-defined hard size requirements from users. Our proposed framework is pluggable and still valid when using the PGM variant to handle the size requirement. Specifically, given a space constraint, the multi-criteria PGM proposes to iteratively estimate the relationship between $\epsilon$ and $size$ with a learnable function $size(\epsilon) = a\epsilon^{-b}$, and automatically outputs the index that minimizes its query time via different estimated $\epsilon$s. Given a size requirement, we can just do the same thing in a dynamic $\epsilon$ scene by setting our $\tilde{\epsilon}$ as $\epsilon$ estimated by the original PGM method.

We present discussions on how the proposed framework can inherit the ability to handle the update case of existing works above. To show how the learned index with dynamic $\epsilon$ performs under different dynamic scenarios, as an example here, we simply replace the *static* $\epsilon$ parameter in the segment-building process of updatable PGM into a *dynamic* one predicted by the proposed $\epsilon$ learner and keep the update processing of PGM unchanged.

We follow the read-heavy and write-heavy workloads setting similar to ALEX Ding et al. (2020), where the workloads are composed of lookup operations and insert operations with different ratios. Specifically, after building learned indexes with a random subset of the whole dataset with the percentage $R_{init} \in (0, 100\%)$, we repetitively perform $N_{lookup}$ random lookup operations and 1 insertion operation in a batch. To simulate different workloads, we set $R_{init}$ to be $[5\%, 10\%, 20\%, 40\%, 60\%, 80\%]$ and $N_{lookup}$ to be $[1, 2, 4, 8, 16]$.

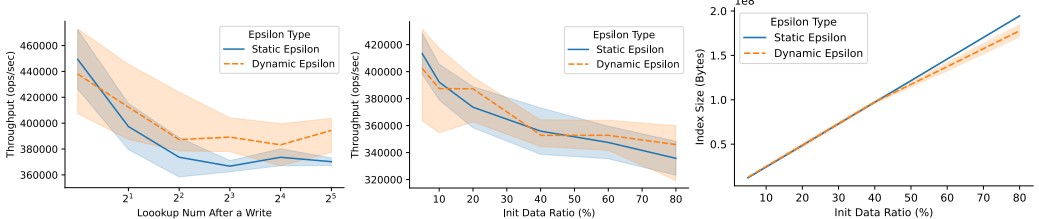

Figure 6: Total throughput (ops/sec) and index size (KB) results of the PGM with static and dynamic $\epsilon$ on two different read-write workloads.

We repeat the experiments 3 times and summarize the total throughput (operations/sec) and index size (KB) in Figure 6. Generally speaking, compared with the PGM with static $\epsilon$, we can observe that PGM with the $\epsilon$ adjustment achieves larger total throughput and smaller sizes in most cases, indicating the effectiveness of the proposed method. We also find that with larger ratios of write (i.e., smaller $N_{lookup}$) and smaller ratios of initial building data ($R_{init}$), the throughput improvements are less and the variances are larger. Note that the current modification with $\epsilon$-learner mostly impacts

the index building stage, and we do not design specific strategies in the insert stage to achieve more precise estimation for the data's local characteristics, which is a promising future direction to further optimize the index performance in update case.

## E  PROOF OF THEOREM 1

Given a learned segment $S_i : y = a_i x + b_i$, denote $c_i$ as the stored position of the last covered data for the $(i-1)$-th segment ($c_1 = 0$ for the first segment). We can write the expectation of $SegErr_i$ for the segment $S_i$ as the following form:

$$\mathbb{E}[SegErr_i] = \mathbb{E}\left[\sum_{j=0}^{(j^*-1)} |a_i X_j + b_i - (j + c_i + 1)|\right],$$

where $j^*$ indicates the length of the segment, and $X_j$ indicates the $j$-th key covered by the segment $S_i$. As studied in Ferragina et al. (2020), the linear-approximation problem with $\epsilon$ guarantee can be modeled as random walk processes. Specifically, $X_j = X_0 + \sum_{k=0}^{j} G_k$ (for $j \in \mathbb{Z}_{>0}$) where $G_k$ is the key increment variable whose mean and variance is $\mu$ and $\sigma^2$ respectively. Denote the $Z_j = X_j - j/a_i + (b_i - c_i - 1)/a_i$ as the $j$-th position of the transformed random walk $\{Z_j\}_{j \in \mathbb{N}}$, and $j^* = \max\{j \in \mathbb{N} | -\epsilon/a_i \leq Z_j \leq \epsilon/a_i\}$ as the random variable indicating the maximal position when the random walk is within the strip of boundary $\pm \epsilon/a_i$. The expectation can be rewritten as:

$$\mathbb{E}\left[\sum_{j=0}^{(j^*-1)} |a_i X_j - j + (b_i - c_i - 1)|\right] = a_i \mathbb{E}\left[\sum_{j=0}^{(j^*-1)} |Z_j|\right]$$

$$= a_i \sum_{n=1}^{\infty} \mathbb{E}\left[\sum_{j=0}^{n-1} |Z_j|\right] \Pr(j^* = n). \tag{3}$$

The last equality in Eq. (3) is due to the definition of expectation. Following the MET algorithm that the $S_i$ goes through the point $(X_0, Y_0 = c_i + 1)$, we get $b_i = -a_i X_0 + c_i + 1$ and we can rewrite $Z_j$ as the following form:

$$Z_0 = 0, \quad Z_j \overset{j \geq 0}{=} X_j - X_0 - j/a_i = \sum_{k=1}^{j} G_k - j/a_i$$

$$= \sum_{k=1}^{j} (G_k - 1/a_i) = \sum_{k=1}^{j} (W_k),$$

where $W_k$ is the walk increment variable of $Z_j$, $\mathbb{E}[W_k] = \mu - 1/a_i$ and $Var[W_k] = \sigma^2$. Under the MET algorithm setting where $a_i = 1/\mu$ and $\varepsilon \gg \sigma/\mu$, the transformed random walk $\{Z_j\}$ has increments with zero mean and variance $\sigma^2$, and many steps are necessary to reach the random walk boundary. With the Central Limit Theorem, we can assume that $Z_j$ follows the normal distribution with mean $\mu_{zj}$ and variance $\sigma_{zj}^2$, and thus $|Z_j|$ follows the folded normal distribution:

$$Z_j \sim \mathcal{N}\big((\mu - 1/a_i)j, j\sigma^2\big),$$
$$\mathbb{E}(|Z_j|) = \mu_{zj}[1 - 2\Phi(-\mu_{zj}/\sigma_{zj})] + \sigma_{zj}\sqrt{2/\pi} \exp(-\mu_{zj}^2/2\sigma_{zj}^2),$$

where $\Phi$ is the normal cumulative distribution function. For the MET algorithm, $a_i = 1/\mu$ and thus the $\mu_{zj} = 0$, $\sigma_{zj} = \sigma\sqrt{j}$, and $\mathbb{E}(|Z_j|) = \sqrt{2/\pi}\sigma\sqrt{j}$. Then the Eq. (3) can be written as

$$\frac{1}{\mu} \sum_{n=1}^{\infty} \mathbb{E}\left[\sum_{j=0}^{n-1} |Z_j|\right] \Pr(j^* = n) < \frac{1}{\mu} \sum_{n=1}^{\infty} \sum_{j=0}^{n-1} \mathbb{E}[|Z_j|] \Pr(j^* = n)$$

$$= \frac{\sigma}{\mu} \sqrt{\frac{2}{\pi}} \sum_{n=1}^{\infty} \sum_{j=0}^{n-1} \sqrt{j} \Pr(j^* = n). \tag{4}$$

For the inner sum term in Eq. (4), we have $(\sum_{j=0}^{n-1}\sqrt{j}) < \frac{2}{3}n\sqrt{n}$ since

$$\sum_{j=0}^{n-1}\sqrt{j} < \sum_{j=0}^{n-1}\sqrt{j} + \frac{\sqrt{n}}{2} < \int_0^n \sqrt{x}\,dx = \frac{2}{3}n\sqrt{n},$$

then the result in Eq. (4) becomes

$$\mathbb{E}[SegErr_i] < \frac{2}{3}\sqrt{\frac{2}{\pi}}\frac{\sigma}{\mu}\sum_{n=1}^{\infty}n\sqrt{n}\Pr(j^*=n)$$

$$= \frac{2}{3}\sqrt{\frac{2}{\pi}}\frac{\sigma}{\mu}\mathbb{E}[(j^*)^{\frac{3}{2}}] = \frac{2}{3}\sqrt{\frac{2}{\pi}}\frac{\sigma}{\mu}\mathbb{E}\left[((j^*)^2)^{\frac{3}{4}}\right]$$

$$\leq \frac{2}{3}\sqrt{\frac{2}{\pi}}\frac{\sigma}{\mu}\left(\mathbb{E}[(j^*)^2]\right)^{\frac{3}{4}},$$

where the last inequality holds due to the Jensen inequality $\mathbb{E}[X^{\frac{3}{4}}] \leq (\mathbb{E}[X])^{\frac{3}{4}}$. Using $\mathbb{E}[j^*] = \frac{\mu^2}{\sigma^2}\epsilon^2$ and $Var[j^*] = \frac{2}{3}\frac{\mu^4}{\sigma^4}\epsilon^4$ derived in MET algorithm Ferragina et al. (2020), we get $\mathbb{E}[(j^*)^2] = \frac{5}{3}\frac{\mu^4}{\sigma^4}\epsilon^4$, which yields the following upper bound:

$$\mathbb{E}[SegErr_i] < \frac{2}{3}\sqrt{\frac{2}{\pi}}(\frac{5}{3})^{\frac{3}{4}}(\frac{\mu}{\sigma})^2\epsilon^3.$$

For the lower bound, applying the triangle inequality into the Eq. (3), we have

$$\frac{1}{\mu}\sum_{n=1}^{\infty}\mathbb{E}\left[\sum_{j=0}^{n-1}|Z_j|\right]\Pr(j^*=n)$$

$$> \frac{1}{\mu}\sum_{n=1}^{\infty}\mathbb{E}\left[|\sum_{j=0}^{n-1}Z_j|\right]\Pr(j^*=n) \tag{5}$$

$$= \frac{1}{\mu}\sum_{n=1}^{\infty}\mathbb{E}[|Z|]\Pr(j^*=n),$$

where $Z = \sum_{j=0}^{n-1}Z_j$. Since $Z_j \sim N(0, \sigma_{zj}^2)$, the $Z$ follows the normal distribution:

$$Z \sim \mathbb{N}\left(\mu_Z = 0,\ \sigma_Z^2 = \sum_{j=0}^{n-1}\sigma_{zj}^2 + \sum_{j=0}^{n-1}\sum_{k=0,k\neq j}^{n-1}r_{jk}\sigma_{zj}\sigma_{zk}\right),$$

where $r_{jk}$ is the correlation between $Z_j$ and $Z_k$. Since $\mu_Z = 0$, the $|Z|$ follows the folded normal distribution with $\mathbb{E}[|Z|] = \sigma_Z\sqrt{2/\pi}$. Since the random walk $\{Z_j\}$ is a process with i.i.d. increments, the correlation $r_{jk} \geq 0$. With $\sigma_{zj} = \sigma\sqrt{j} > 0$ and $r_{jk} \geq 0$, we have

$$\mathbb{E}[|Z|] > \sqrt{\frac{2}{\pi}}\sum_{j=0}^{n-1}\sigma_{zj} > \sigma\sqrt{n(n-1)/\pi} > \frac{\sigma(n-1)}{\sqrt{\pi}},$$

and the result in Eq. (5) becomes:

$$\mathbb{E}[SegErr_i] > \frac{1}{\mu}\sum_{n=1}^{\infty}\mathbb{E}\left[|\sum_{j=0}^{n-1}Z_j|\right]\Pr(j^*=n)$$

$$> \frac{\sigma}{\mu}\sqrt{\frac{1}{\pi}}\sum_{n=1}^{\infty}(n-1)\Pr(j^*=n)$$

$$= \frac{\sigma}{\mu}\sqrt{\frac{1}{\pi}}\mathbb{E}[j^*-1] = \sqrt{\frac{1}{\pi}}(\frac{\mu}{\sigma}\epsilon^2 - \frac{\sigma}{\mu}).$$

Since $\epsilon \gg \frac{\sigma}{\mu}$, we can omit the right term $\sqrt{\frac{1}{\pi}}\frac{\sigma}{\mu}$ and finish the proof.

## F    Learned Slopes of Other $\epsilon$-Bounded Methods

As shown in Theorem 1, we have known how $\epsilon$ impacts the $SegErr_i$ of each segment learned by the MET algorithm, where the theoretical derivations largely rely on the slope condition $a_i = 1/\mu$. Here we prove that for other $\epsilon$-bounded methods, the learned slope of each segment (*i.e.*, $a_i$ of $S_i$) concentrates on the reciprocal of the expected key interval as shown in the following Theorem.

**Theorem 2.** *Given an $\epsilon \in \mathbb{Z}_{>1}$ and an $\epsilon$-bounded learned index algorithm $\mathcal{A}$. For a linear segment $S_i : y = a_i x + b_i$ learned by $\mathcal{A}$, denote its covered data and the number of covered keys as $\mathcal{D}_i$ and $Len(\mathcal{D}_i)$ respectively. Assuming the expected key interval of $\mathcal{D}_i$ is $\mu_i$, the learned slope $a_i$ concentrates on $\tilde{a} = 1/\mu_i$ with bounded relative difference:*

$$(1 - \frac{2\epsilon}{\mathbb{E}[Len(\mathcal{D}_i)] - 1})\tilde{a} \quad \leq \quad E[a_i] \quad \leq \quad (1 + \frac{2\epsilon}{\mathbb{E}[Len(\mathcal{D}_i)] - 1})\tilde{a}.$$

*Proof.* For the learned linear segment $S_i$, denote its first predicted position and last predicted position as $y'_0$ and $y'_n$ respectively, we have its slope $a_i = \frac{y'_n - y'_0}{x_n - x_0}$. Notice that $y_0 - \epsilon \leq y'_0 \leq y_0 + \epsilon$ and $y_n - \epsilon \leq y'_n \leq y_n + \epsilon$ due to the $\epsilon$ guarantee, we have $y_n - y_0 - 2\epsilon \leq y'_n - y'_0 \leq y_n - y_0 + 2\epsilon$ and the expectation of $a_i$ can be written as

$$\mathbb{E}[\frac{y_n - y_0 + 2\epsilon}{x_n - x_0}] \quad \leq \quad E[a_i] = \frac{y'_n - y'_0}{x_n - x_0} \quad \leq \quad \mathbb{E}[\frac{y_n - y_0 + 2\epsilon}{x_n - x_0}].$$

Note that for any learned segment $S_i$ whose first covered data is $(x_0, y_0)$ and last covered data is $(x_n, y_n)$, we have $\mathbb{E}[\frac{x_n - x_0}{y_n - y_0}] = \mu_i$ and thus the inequalities become

$$\frac{1}{\mu} - \mathbb{E}[\frac{2\epsilon}{x_n - x_0}] \quad \leq \quad E[a_i] \quad \leq \quad \frac{1}{\mu} + \mathbb{E}[\frac{2\epsilon}{x_n - x_0}].$$

Since $\tilde{a} = 1/\mu_i$ and $\mathbb{E}[x_n - x_0] = (\mathbb{E}[Len(\mathcal{D}_i)] - 1)\mu_i$, we finish the proof. $\qquad\square$

The Theorem 2 shows that the relative deviations between learned slope $a_i$ and $\tilde{a}$ are within $2\epsilon/(\mathbb{E}[Len(\mathcal{D}_i)] - 1)$. For the MET and PGM learned index methods, we have the following corollary that depicts more precise deviations without the expectation term $\mathbb{E}[Len(\mathcal{D}_i)]$.

**Corollary 2.1.** *For the MET method Ferragina et al. (2020) and the optimal $\epsilon$-bounded linear approximation method that learns the largest segment length used in PGM Ferragina & Vinciguerra (2020b), the slope relative differences are at $O(1/\epsilon)$.*

*Proof.* We note that the segment length of a learned segment is at $O(\epsilon^2)$ for the MET algorithm, which is proved in the Theorem 1 of Ferragina et al. (2020). Since PGM achieves the largest learned segment length that is larger than the one of the MET algorithm, we finish the proof. $\qquad\square$

## G    The Algorithm of Dynamic $\epsilon$ Adjustment

We summarize the proposed algorithm below. In Section 3.4, we provide detailed descriptions of the initialization and adjustment sub-procedures. The $lookahead()$ and $optimize()$ are in the "**Look-ahead Data**" and "$\widetilde{SegErr}$ **and Optimization**" paragraph respectively.

## H    Implementation Details

**Baselines.**    All the experiments are conducted on a Linux server with an Intel Xeon Platinum 8163 2.50GHz CPU. We first introduce more details and the implementation of baseline learned index methods. *MET* (Ferragina et al., 2020) fixes the segment slope as the reciprocal of the expected key interval, and goes through the first available data point for each segment. *FITing-Tree* (Galakatos et al., 2019) adopts a greedy shrinking cone algorithm and the learned segments are organized with a B$^+$-tree. Here we use the stx::btree (v0.9) implementation (Bingmann, 2013) and set the filling factors of inner nodes and leaf nodes as $100\%$, *i.e.*, we adopt the full-paged filling manner. *Radix-Spline* (Kipf et al., 2020) adopts a greedy spline interpolating algorithm to learn spline points,

---

**Algorithm** Dynamic $\epsilon$ Adjustment with Pluggable $\epsilon$ Learner

---

**Input:** $\mathcal{D}$: Data to be indexed, $\mathcal{A}$: Learned index algorithm, $\tilde{\epsilon}$: Expected $\epsilon$, $\rho$: Length percentage for look-ahead data

**Output:** **S**: Learned segments with varied $\epsilon$s

1: initial parameters $w_{1,2,3}$ of the learned function: $f(\epsilon, \mu, \sigma) = w_1(\frac{\mu}{\sigma})^{w_2}\tilde{\epsilon}^{w_3}$
2: initial mean length of learned segments so far: $Len(\mathcal{D_S}) \leftarrow 404$
3: $\mathbf{S} \leftarrow \varnothing,\ \ (\hat{\mu}/\hat{\sigma}) \leftarrow 0$
4: **repeat**
5:   Get data statistic:
6:    $(\mu/\sigma) \leftarrow lookahead(\mathcal{D}, Len(\mathcal{D_S}) \cdot \rho)$
7:   Adjust $\epsilon$ based on the learner:
8:    $\epsilon^* \leftarrow \left( \widetilde{SegErr}/w_1(\frac{\mu}{\sigma})^{w_2} \right)^{1/w_3}$
9:   Learn new segment $S_i$ using adjusted $\epsilon^*$:
10:    $[\mathcal{S}_i, \mathcal{D}_i] \leftarrow \mathcal{A}(\mathcal{D}, \epsilon^*)$
11:    $\mathbf{S} \leftarrow \mathbf{S} \cup \mathcal{S}_i$
12:    $\mathcal{D} \leftarrow \mathcal{D} \setminus \mathcal{D}_i,\ \ \mathcal{D_S} \leftarrow \mathcal{D_S} \cup \mathcal{D}_i$
13:   Online update $Len(\mathcal{D_S})$:
14:    $Len(\mathcal{D_S}) \leftarrow$ running-mean$\big(Len(\mathcal{D_S}), Len(\mathcal{D}_i)\big)$
15:    $(\hat{\mu}/\hat{\sigma}) \leftarrow$ running-mean$\big((\hat{\mu}/\hat{\sigma}), (\mu/\sigma)\big)$
16:   Train the learner with ground truth:
17:    $w_{1,2,3} \leftarrow optimize(f, S_i, SegErr_i)$
18:    $\widetilde{SegErr} \leftarrow w_1(\hat{\mu}/\hat{\sigma})^{w_2}\tilde{\epsilon}^{w_3}$
19: **until** $\mathcal{D} = \varnothing$

---

and the learned spline segments are organized with a flat radix table. We set the number of radix bits as $r = 16$ for the Radix-Spline method, which means that the leveraged radix table contains $2^{16}$ entries. *PGM* (Ferragina & Vinciguerra, 2020b) adopts a convex hull based algorithm to achieve the minimum number of learned segments, and the segments can be organized with the help of binary search, CSS-Tree (Rao & Ross, 1999) and recursive structure. Here we implement the recursive version since it beats the other two variants in terms of indexing performance. For all the baselines and our method, we adopt exponential search to better leverage the predictive models since the query complexity using exponential search corresponds to the preciseness of models (*MAE*) as we analyzed in Appendix B.

**Evaluation Metrics.** We evaluate the index performance in terms of its size, prediction preciseness, and total querying time. Specifically, we report the number of learned segments $N$, the index size in bytes, the *MAE* as $\frac{1}{|D|}\sum_{(x,y) \in D}|y - \mathbf{S}(x)|$, and the total querying time per query in ns (*i.e.*, we perform querying operations for all the indexed data, record the total time of getting the payloads given the keys, and report the time that is averaged over all the queries). For a quantitative comparison w.r.t. the trade-off improvements, we calculate the **A**rea **U**nder the **N**-*MAE* **C**urve (AUNEC) where the x-axis and y-axis indicate $N$ and *MAE* respectively. For the AUNEC metric, the smaller, the better.

**Hyper-parameters.** We describe a few additional details of the proposed framework in terms of the $\epsilon$-learner initialization and the hyper-parameter setting. For the $w_{1,2,3}$ of the $\epsilon$-learner shown in the Eq. (2), We empirically found that this light-weight initialization leads to better index performance compared to the versions with random parameter initialization, and it benefits the exploration of diverse $\epsilon^*$, *i.e.*, leading to the larger variance of the dynamic $\epsilon$ sequence $[\epsilon_1, \ldots, \epsilon_i, \ldots, \epsilon_N]$. As for the hyper-parameter $\rho$ (described in the Section 3.4), we conduct a grid search over $\rho \in [0.1, 0.4, 0.7, 1.0]$ on Map and IoT datasets. We found that all the $\rho$s achieve better $N$-*MAE* trade-off (*i.e.*, smaller AUNEC results) than the fixed $\epsilon$ versions. Since the setting $\rho = 0.4$ achieves averagely best results on the two datasets, we set $\rho$ to be $0.4$ for the other datasets.

# I   DATASET DETAILS

Our framework is verified on several widely adopted datasets having different data scales and distributions. *Weblogs* Kraska et al. (2018); Galakatos et al. (2019); Ferragina & Vinciguerra (2020b) contains about 715M log entries for the requests to a university web server and the keys are log timestamps. *IoT* Galakatos et al. (2019); Ferragina & Vinciguerra (2020b) contains about 26M event entries from different IoT sensors in a building and the keys are recording timestamps. *Map* dataset Kraska et al. (2018); Galakatos et al. (2019); Ding et al. (2020); Ferragina & Vinciguerra (2020b); Li et al. (2021b) contains location coordinates of 200M places that are collected around the world from the Open Street Map OpenStreetMap contributors (2017), and the keys are the longitudes of these places. *Lognormal* Ferragina & Vinciguerra (2020b) is a synthetic dataset whose key intervals follow the lognormal distribution: $ln(G_i) \sim \mathcal{N}(\mu_{lg}, \sigma_{lg}^2)$. To simulate the varied data characteristics among different localities. We generate 20M keys with 40 partitions by setting $\mu_{lg} = 1$ and setting $\sigma_{lg}$ with a random number within $[0.1, 1]$ for each partition.

We normalize the positions of stored data into the range $[0, 1]$, and thus the key-position distribution can be modeled as a Cumulative Distribution Function (CDF). We plot the CDFs and zoomed-in CDFs of experimental datasets in Figure 7 and Figure 8 respectively, which intuitively illustrate the diversity of the adopted datasets. For example, the CDF visualization of the Map dataset shows that it has a fairly shifted distribution across different data localities, verifying of the necessity of dynamically adapting and adjusting the learned index algorithms just as we considered in this paper.

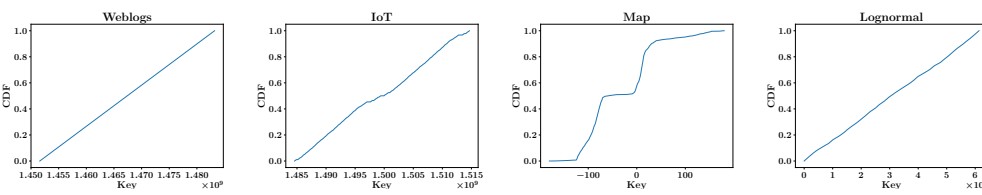

Figure 7: CDFs of adopted datasets.

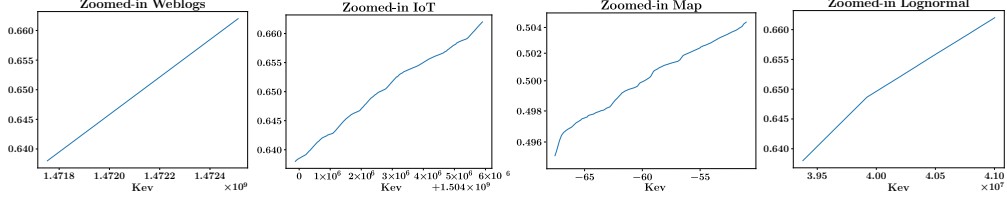

Figure 8: Zoomed-in CDFs of adopted datasets.

# J   ADDITIONAL EXPERIMENTAL RESULTS

## J.1   OVERALL INDEX PERFORMANCE.

For the $N$-*MAE* trade-off improvements and the actual querying efficiency improvements brought by the proposed framework, we illustrate more $N$-*MAE* trade-off curves in Figure 9 and querying time results in Figure 10. We also mark the 99th percentile (P99) latency as the right bar, which is a useful metric in industrial-scale practical systems. Recall that the $N$-*MAE* trade-off curve adequately reflects the *index size* and *querying time*: (1) the *segment size in bytes* and $N$ are only different by a constant factor, e.g., the size of a segment can be 128bit if it consists of two double-precision float parameters (slope and intercept); (2) the querying operation can be done in $O(log(N) + log(|y - \hat{y}|))$ as we mentioned in Section 3.1, thus a better $N$-*MAE* trade-off indicates a better querying efficiency. From these figures, we can see that the dynamic $\epsilon$ versions of all the baseline methods achieve better $N$-*MAE* trade-off and better querying efficiency, verifying the effectiveness and the wide applicability of the proposed framework. Regards the p99 metrics, we can see that the dynamic version achieves comparable or even better P99 results than the static version, showing that the proposed method not

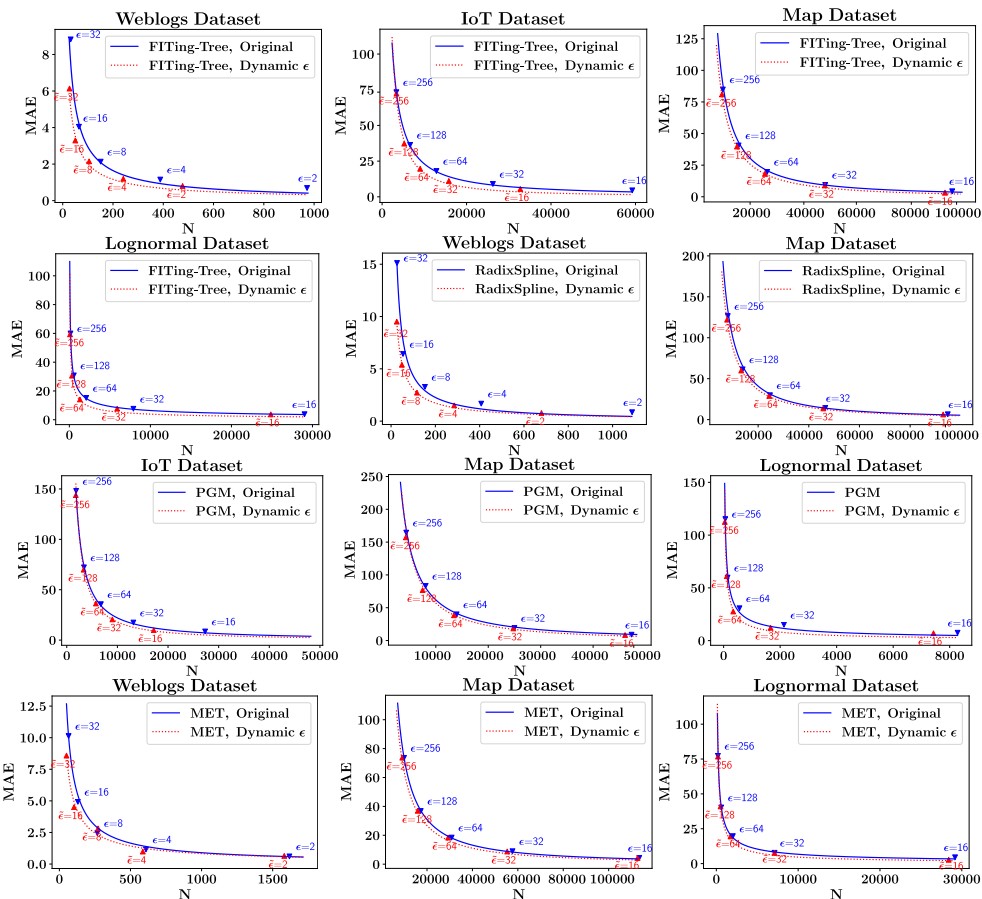

Figure 9: The additional $N$-*MAE* trade-off curves for learned index methods.

only improves the average lookup time, but also has good robustness. This is because our method can effectively adjust $\epsilon$ based on the expected $\tilde{\epsilon}$ and data characteristic, making the $\{\epsilon_i\}$ fluctuated within a moderate range.

## J.2 ABLATION STUDY

To examine the necessity and the effectiveness of the proposed framework, in Section 4.3, we compare the proposed framework with three dynamic $\epsilon$ variants for the FITing-Tree method. Here we demonstrate the AUNEC relative changes for the Radix-Spline method with the same three variants in Table 6 and similar conclusions can be drawn.

Table 6: The AUNEC relative changes of dynamic $\epsilon$ variants compared to the Radix-Spline method with the proposed framework.

|  | Random $\epsilon$ | Polynomial Learner | Least Square Learner |
| --- | --- | --- | --- |
| Weblogs | +81.23% | +56.20% | -9.56% |
| IoT | +74.78% | +53.28% | +9.81% |
| Map | +60.67% | +7.34% | +0.45% |
| Lognormal | +83.16% | +55.01% | $-11.23\%$ |
| Average | +74.96% | +42.96% | $-2.63\%$ |

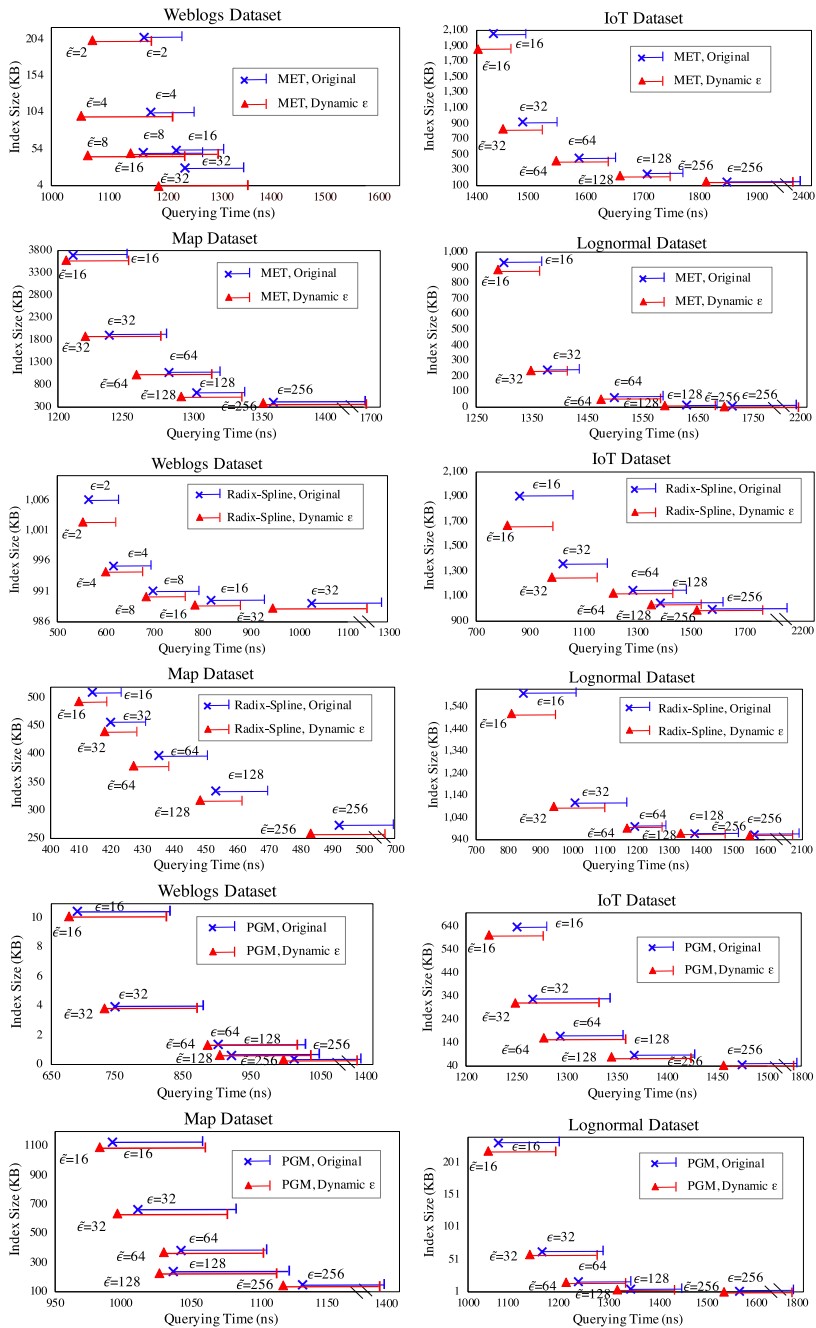

Figure 10: Improvements in terms of querying times for learned index methods with dynamic $\epsilon$.

## J.3 THEORETICAL RESULTS VALIDATION

We study the impact of $\epsilon$ on $SegErr_i$ for the MET algorithm in Theorem 1, where the derivations are based on the setting of the slope condition $a_i = 1/\mu$. To confirm that the proposed framework also works well with other $\epsilon$-bounded learned index methods, we analyze the learned slopes of other $\epsilon$-bounded methods in Theorem 2. In summary, we prove that for a segment $S_i : y = a_i x + b_i$ whose covered data is $\mathcal{D}_i$ and the expected key interval of $\mathcal{D}_i$ is $\mu_i$, then $a_i$ concentrates on $1/\mu_i$ within $2\epsilon/(\mathbb{E}[Len(\mathcal{D}_i)] - 1)$ relative deviations. Here we plot the learned slopes of baseline learned index methods in Figure 11 (Map dataset) and in Figure 13 (IoT, Weblogs and Lognormal datasets). We can see that the learned slopes of other methods indeed center along the line $a_i = 1/\mu_i$, showing the

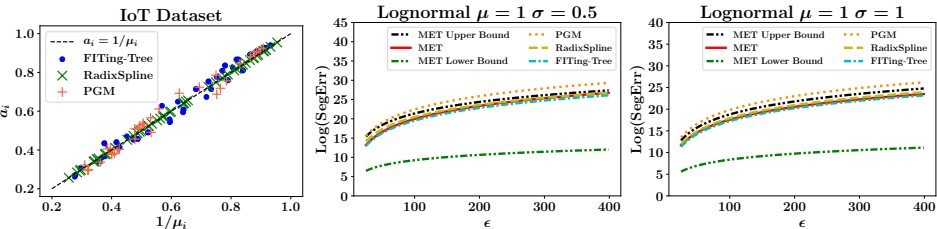

Figure 11: Learned slopes.       Figure 12: Illustration of the derived bounds.

close connections among these methods and confirming that the proposed framework can work well with other $\epsilon$-bounded learned index methods.

We further compare the theoretical bounds with the actual $SegErr_i$ for all the adopted learned index methods. We show the results on the Lognormal dataset in Figure 12, and the results on another two datasets *Gamma* and *Uniform* in Figure 14, where the key intervals of the latter two datasets follow gamma distribution and uniform distribution respectively. As expected, we can see that the MET method has the actual $SegErr_i$ within the derived bounds, verifying the correctness of the Theorem 1. Besides, the other $\epsilon$-bounded methods show the same trends with the MET method, providing evidence that these methods have the same mathematical forms as we derived, and thus the $\epsilon$-learner also works well with them.

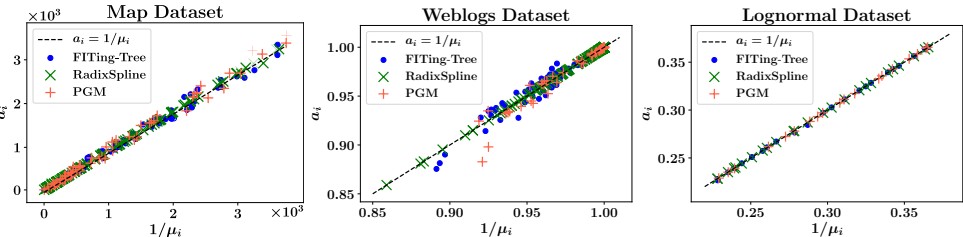

Figure 13: Learned slopes on the IoT, Weblogs and Lognormal datasets.

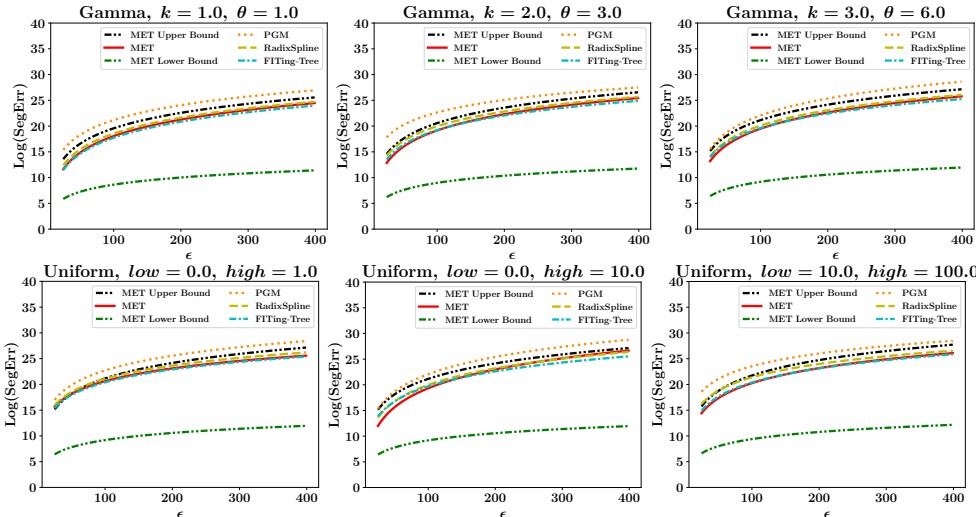

Figure 14: Illustrations of the derived bounds on *Gamma* and *Uniform* datasets.

## J.4  CV as an Indicative Quantity

The *coefficient of variation* (CV) value, *i.e.*, $CV = \sigma/\mu$, plays an important factor in our bounds to reflect the linearity degree of the data. We have seen that $CV$ is effective to help dynamically adjust $\epsilon$ in our framework as shown in our experiments. Here we explore that *whether the CV value can be an indicative quantity to shed light on what types of data will benefit from our dynamic adjustment.* To be specific, we calculate the $CV$ values of the experimental datasets and compare them with the trade-off improvements.

The global $CV$ values of IoT, Map, Lognormal, and Weblogs are 65.24, 11.12, 0.85, and 0.013 respectively, while their AUNEC improved by 20.71%, 6.47%, 21.89%, and 26.96% respectively. With the exception of IoT, the rest of the results show that *the smaller the $CV$ value is, the greater the trade-off improvement of dynamic $\epsilon$ brings*. We find that IoT is a locally linear but globally fluctuant dataset. We then divide the data into 5000 segments and calculate their average $CV$ values. The local $CV$ values of IoT, Map, Lognormal, and Weblogs are 0.95, 2.18, 0.63, and 0.005 respectively, which is consistent with the improvement trends. Intuitively, when the local $CV$ value is small, the local data is hard-to-fit with a few linear segments if we adopt an improper $\epsilon$, and we need more fine-grained $\epsilon$ adjustment rather than the fixed setting. Thus we can expect more performance improvements in this case. The calculation of actual $CV$ values of real-world datasets helps to validate our $\epsilon$ analysis based on the $CV$ values, and provides further insight into the scenarios where the proposed method has strong potential to outperform existing methods.

