# OpenReview forum: "Learned Index with Dynamic $\epsilon$"
_ICLR.cc/2023/Conference — ICLR 2023 poster_

### Official Review · Reviewer_apVD · 2022-10-24

**Confidence:** 3
**Correctness:** 4
**Technical Novelty And Significance:** 2
**Empirical Novelty And Significance:** 2
**Recommendation:** 5

**Clarity, Quality, Novelty And Reproducibility:**

This paper has a lot of formula derivation, which provides theoretical basis for this paper. This paper is well organized and well structured, but the method proposed is not innovative enough.

**Strength And Weaknesses:**

Strength
1. A large number of formulas are derived in this paper, which provide theoretical basis for the proposed method.
2. The comparative experiments in this paper are very rich, and several basic metrics of learned index are measured.
3. Whether the method proposed in this paper is universal, that is, it performs well under any data distribution.

Weaknesses
1. This paper uses a large number of mathematical symbols and formulas, it is recommended to use a table to summarize for the convenience of readers.
2. It is recommended to use examples in complex part, such as the flow of the entire framework.
3. This paper addresses an existing problem, and it suggests implement other methods based on dynamic ε in experience.
4. This paper has carried out several groups of comparative experiments, but the experimental results are not outstanding.
5. There are some grammatical errors such as no subject.


**Summary Of The Paper:**

This paper solves the problem that existing learned index methods adopt a fixed ε for all the learned segments, which neglecting the diverse characteristics of different data localities and propose a mathematically-grounded learned index framework with dynamic ε, which is efficient and pluggable to existing learned index methods.

**Summary Of The Review:**

This paper addresses an existing problem, but the solution is not innovative enough and does not perform much better.

---

> ### Author Response · Authors · 2022-11-19
> **Responses to Reviewer apVD**
>
> Many thanks for the insightful and helpful comments by the reviewer! We make the following responses point by point to address your comments:
>
> **W1: "It is recommended to use a table to summarize the mathematical symbols and formulas."**
>
> Thanks for your helpful suggestion! To make the reading more convenient, we added a notation table accordingly in the revision (Appendix A, Table 4).
>
> **W2: "It is recommended to use examples in complex part, such as the flow of the entire framework."**
>
> Thank you very much for your suggestion! We presented step-by-step descriptions for the overview framework (Figure 1 and Sec 3.2), and also provided a case study to illustrate the dynamic learning process (Figure 4).
> As a more detailed complementation to Figure 1, we newly add a running example to show the step-by-step workflow of the $\epsilon$-bounded learned index in Figure 5. Thanks again!
>
> **W3: "This paper addresses an existing problem, and it suggests implement other methods based on dynamic $\epsilon$ in experience."**
>
> 1. [**Novel methodology and theoretical analysis**] Many existing $\epsilon$-bounded learned index works [1,2,3,4] focus on the mechanism of modeling linear segments, and some of them re-used previous mathematical results such as [2].
> Another related work, MET [5], mainly focuses on theoretical research with preliminary experiments on synthetic datasets. Compared with these works, we not only propose a new learned index algorithm, but also give new theoretical results, which can be regarded as a double innovation.
> 2. [**The benefits of pluggability**] We propose a pluggable framework for enhancing existing $\epsilon$-bounded learned index methods from a new perspective. The pluggable framework provides convenience and wide applicability, which means that we don't need to change the code a lot when using it, and it can be applied to a class of $\epsilon$-bounded algorithms as shown in the experiments where we took four SOTA methods as the base methods to be improved.
> 2. [**A new perspective and setting**] To our knowledge, no prior learned index work has investigated the dynamic $\epsilon$ setting considering the diversity of different data localities. The proposed dynamic adjustment of $\epsilon$ demonstrates the feasibility of *learning to leverage the distribution information of data localities*, as shown in our mathematical derivation linking the indexing performance, $\epsilon$ and data statistics $\mu/\sigma$ ( Section 3.3) and the proposed $\epsilon$-learner (Section 3.4).
> We believe this paper will contribute a deeper understanding of how the $\epsilon$ and characteristics of data localities impact the indexing performance trade-off, and provide new insight into effective index learning with fine-grained adjustment.
>
> [1] Galakatos, Alex, et al. "Fiting-tree: A data-aware index structure." SIGMOD, 2019.
>
> [2] Ferragina, Paolo, and Giorgio Vinciguerra. "The PGM-index: a fully-dynamic compressed learned index with provable worst-case bounds." VLDB, 2020.
>
> [3] Kipf, Andreas, et al. "RadixSpline: a single-pass learned index." aiDM, 2020.
>
> [4] Li, Pengfei, et al. "A scalable learned index scheme in storage systems." arXiv, 2019.
>
> [5] Ferragina, Paolo, Fabrizio Lillo, and Giorgio Vinciguerra. "Why Are Learned Indexes So Effective?." ICML, 2020.
>
> **W4:  "The experimental results are not outstanding."**
>
> 1. Although the maximum *absolute* improvements in the index size and query time are about 1MB and 100ns respectively, the *relative* improvements are significant by up to about 82% and 12.9% respectively (FITing-Tree on IoT, $\epsilon$=16, Fig. 3).
> 2. Note that the reported query time corresponds *a single query*, and the proposed algorithm framework *retains the online learning manner* (with linear complexity). This means that we can scale to very large datasets and achieve considerable improvements in terms of the total query time.
> 3. Taking PGM-Index as a more specific example, for real datasets, it improves index size by up to 63% over its previous baseline, FITing-Tree, while the improvement in query performance is unclear and not reported in their paper. By contrast, we achieve larger relative improvements than the published SOTA method.
>
> **W5: "There are some grammatical errors such as no subject."**
>
> We did several more careful proofreads and modified a few grammatical errors. Please point out the specific sentences if you don't mind, and we will correct them in time. Thanks again!

---

### Official Review · Reviewer_QKCD · 2022-10-24

**Confidence:** 4
**Clarity, Quality, Novelty And Reproducibility:** The paper is well written and easy to…
**Correctness:** 3
**Technical Novelty And Significance:** 3
**Empirical Novelty And Significance:** 3
**Recommendation:** 6

**Strength And Weaknesses:**

Strength
1. The paper present the comprehensive theoretical analysis and corresponding proofs.
2. The proposed methods are easy to applied to all piece-wise-approximation-based learned index.

Weakness:
1. It would be clearer for readers to understand the learned index when discuss about the related work on the learned index from different optimization aspect, such as epsilon-bounded mechanism, index structure. New works should be also discussed here, such as LIPP [1] NFL [2], CARMI [3], FINEdex [4].

2. The paper claims that the proposed framework has the ability to handle dynamic udpate case (Section 3.2) and the cost of the estimation for the epsilon is only paied once (Section 4.2 Indexing Building Cost). However, the experiments in the Section 4 or Appendix only includes the results on read-only workloads. The results on read-write workloads are missing. The update issues in indexing are complicated and includes several aspects of research directions. For example, to support updates, existing indexes such as LIPP [1], Alex, PGM often propose new operations to adjust the inner structure which may involve rebuilding parts of the structure. From this perspective, does the previous epsilons still fits? Or does the index need to pay the estimation cost again?

3. The codes are incomplete and are all python codes. All existing learned index are implemented in C++ for obtaining the faster performance. The codes for implementing the framework or measuring the index's performance (latency or index size) are missing.

[1] Jiacheng Wu, Yong Zhang, Shimin Chen, Yu Chen, Jin Wang, Chunxiao Xing. Updatable Learned Index with Precise Positions. Proc. VLDB Endow. 14(8): 1276-1288 (2021)
[2] Shangyu Wu, Yufei Cui, Jinghuan Yu, Xuan Sun, Tei-Wei Kuo, Chun Jason Xue. NFL: Robust Learned Index via Distribution Transformation. Proc. VLDB Endow. 15(10): 2188-2200 (2022)
[3] Jiaoyi Zhang, Yihan Gao. CARMI: A Cache-Aware Learned Index with a Cost-based Construction Algorithm. Proc. VLDB Endow. 15(11): 2679-2691 (2022)
[4] Pengfei Li, Yu Hua, Jingnan Jia, Pengfei Zuo. FINEdex: A Fine-grained Learned Index Scheme for Scalable and Concurrent Memory Systems. Proc. VLDB Endow. 15(2): 321-334 (2021)

**Summary Of The Paper:**

Recent learned indexes leverage the machine learning models to replace the traditional index structures to achieve a superior performance. However, existing learned indexes adopt a fixed prediction error threshold to build the index. This paper focus on the diverse characteristics of different data localities and propose a pluggable learned index framework to dynamically adjust the prediction error threshold. The experiments shows the proposed methods can achieve a better trade-off between space and time effiency.

**Summary Of The Review:**

The paper proposes useful method for adaptive epsilon for learned index. The update is an important consideration for database. Before seeing more evaluations and comparisons, I would recommend a borderline score.

---

> ### Author Response · Authors · 2022-11-19
> **Responses to Reviewer QKCD**
>
> Thank you very much for your appreciation and detailed comments! We make the following responses point by point to address your comments:
>
> **W1: "It would be clearer for readers to understand the learned index when discuss about the related work on the learned index from different optimization aspects."**
>
> Many thanks for your constructive suggestion!
> - Due to the space limitation, we did not discuss these works in the main content as we focus on the $\epsilon$-bounded methods.
> In Appendix K, we presented a more detailed comparison of the data-layout-optimization-based methods. Besides the advantage of worst-case guarantee with $\epsilon$, we empirically show that the learned index with dynamic $\epsilon$ can achieve comparable query time and smaller index size than the baseline method ALEX. We also pointed out that although the data layout optimization goes beyond our scope, our insight can contribute to the data-layout-optimization-based methods that introduce hyper-parameters in local linear segments learning, since *the data distribution can be varied across different localities* as we have shown in experiments.
> - In order to give readers a more complete and clear understanding of the learned index methods, we have now introduced the learned index methods based on the other optimization aspect in a more early position (the background section), including the four works you mentioned. Thanks again!
>
>
> **W2: "The results on read-write workloads are missing. The update issues in indexing are complicated and includes several aspects of research directions. ... From this perspective, does the previous epsilons still fits?"**
>
> Thanks for your insightful question!
> - We agree that "The update issues in indexing are complicated and include several aspects of research directions." Since this is the first paper discussing dynamic $\epsilon$ setting and has already conveyed a lot of content (20+ pages), in Appendix G, we gave some discussion on how the proposed framework can inherit the ability to handle update cases of existing works and did not provide the results on read-write workloads.
> - Within the response stage, we newly conduct experiments to examine the performance of the PGM index with static and dynamic $\epsilon$ under different read-write workloads. The detailed setting, results, and discussions can be found in the revision (Appendix J.5 and Figure 14). To summarize, the PGM with dynamic $\epsilon$ gains larger improve the throughput of PGM with static $\epsilon$, showing the effectiveness and the extensibility of the proposed framework in the update case.
>
>
> **W3: The codes are incomplete and are all python codes.**
>
> Thank you for the comments! We have uploaded the c++ code in the anonymous [link](https://github.com/AnonyResearcher/ICLR-2240). In `index_dynamic_epsilon.cpp`, we call the "evaluate_indexer" function of different learned indexes, e.g., the detailed implementation can be found in *include/radix_spline/RadixSplineIndexer.h, lines 204 ~ 275.*,
> *include/radix_spline/METIndexer.h, lines 264 ~ 335.*, and
> *include/radix_spline/PgmIndexer.h, lines 152 ~ 217*. In *benchmark/bench_dynamic_epsilon.cpp*, we call "benchmark_workloads" function to conduct the evaluation.

---

> > ### Comment · Reviewer_QKCD · 2022-12-08
> > **Thanks for the clarification.**
> >
> > Thanks for the detailed clarification that addresses my concerns. I would keep the positive score with a high confidence.

---

### Official Review · Reviewer_WrZf · 2022-10-25

**Confidence:** 2
**Clarity, Quality, Novelty And Reproducibility:** See the last question for details.
**Correctness:** 3
**Technical Novelty And Significance:** 3
**Empirical Novelty And Significance:** 3
**Recommendation:** 6

**Strength And Weaknesses:**

Strength

1. The paper gives a deep understanding of the learned index problem and first uses a dynamic $\epsilon$, which is interesting to me.
2. The experimental result shows the advantage of the proposed algorithm.

Weaknesses

1. It is unclear to me why the theory results in this paper show that the dynamic $\epsilon$ can be better than the previous results. Can the author explain that?
2. I am not very familiar with the literature on leaned indexing, and I feel that the discussion of the literature is a little hard for me to follow. Maybe the authors can add more explanation in the next version. Also, the authors give some detailed explanations in the appendix, however, some of them seem not to be referred in the main content.

**Summary Of The Paper:**

The paper studies the learned index problem and proposes a new approach. Unlike the previous approaches that fix the prediction error guarantee $\epsilon$, the proposed approach dynamically adjusts the $\\epsilon$.  The paper also gives the corresponding theoretical analysis. Experiments also demonstrate the advantage of the proposed algorithm.

**Summary Of The Review:**

Overall I think it is an interesting paper. However, I feel like the presentation of the paper can be further improved. I am not very familiar with the literature on the learned indexing problem and I am open to hearing feedback from other reviewers.

---

> ### Author Response · Authors · 2022-11-19
> **Responses to Reviewer WrZf**
>
> Many thanks for your appreciation and insightful comments! We make the following responses point by point to address your comments:
>
> **W1: Why do the theory results show the dynamic $\epsilon$ can be better?**
>
> Thanks for your question!
> - In Section 3.1, we first discuss the intuition and rationale of how the indexing **performance trade-off** between index size and querying time can be **reflected using the quantity $SegErr$**, and **why adjusting $\epsilon$ can impact $SegErr$.**
> - With this foundation in mind, in Section 3.3, we then propose Theorem 1 to make a more precise quantitative analysis of the relationship among $SegErr$, $\epsilon$, and data characteristics $\mu,\sigma$.
> The theoretical results show double advantages:
>     - on the one hand, the derived closed-form relationships can be used to guide the training of the $\epsilon$-learners's introduced in Equation (2). The formulation indicates that we can train the $\epsilon$-learner to predict suitable $\epsilon$ for different data local characteristics, and to achieve a better performance trade-off (smaller $SegErr$).
>     - on the other hand, we get a tighter bound than the trivial bound derived by the results of MET, which can help us to make the training of $f$ more efficient with smaller hypotheses space to be searched.
>
> **W2: The discussion of the literature is a little hard for me to follow.**
>
> Thank you for pointing out that some readers might be unfamiliar with the context of learned index! In Section 2, we briefly introduced the problem settings for learned index and summarized several $\epsilon$-bounded learned index methods which are most related to our work. To explain the related work more clearly, we add a running example to show the step-by-step workflow of the $\epsilon$-bounded learned index in Figure 5. Thanks again!

---

### Author Response · Authors · 2022-12-05
**Gentle reminder of the discussion deadline**

Dear AC and Reviewers,

We sincerely thank you for your time and efforts in reviewing our paper as well as the constructive and insightful feedback! During the response period, we carefully read and responded to all comments from the reviewers, which can be summarized as follows:

- [Experiment] for the update scenario in learned index (QKCD), we add more discussion and conduct new experiments to show the extensibility and effectiveness of the proposed framework under different read-write workloads.
- [Readability] for the background of the learned index (WrZf, apVD, & QKCD), we provide a new running example to show the step-by-step workflow of learned index, and more descriptions about the related works; for the a large number of used mathematical symbols and formulas (apVD), we add a summarized table.
- [Clarification] for the question about how to use the theory results (WrZf), and comments about the contribution of the paper (apVD), we provide more structured and detailed explanation.

We believe the paper has been further improved with the your helpful comments, and hope that these responses will convince you to lean more toward acceptance of the paper.

The discussion phase will end in a few days, and we have received no responses. Please let us know if you have any additional questions or suggestions. We are happy to engage more and address them, thanks again!


Best, authors

---

### Decision · Program_Chairs · 2023-01-20

**Decision:**

Accept: poster

**Justification For Why Not Higher Score:**

The writing can still be improved, especially by showing more examples and presenting an abbreviated version of the pseudo-code in Appendix F.

**Justification For Why Not Lower Score:**

The paper addresses an important problem, and the method appears effective both in theory and practice.

**Metareview: Summary, Strengths And Weaknesses:**

This paper addresses learning index for database by uncovering hidden data distribution and guaranteeing bounded prediction error $\epsilon$.  The key novelty is to use a dynamic value of $\epsilon$ for learned segments, adapting to the data localities. It is efficient and readily integrates to existing index methods.  The link between $\epsilon$ and data characteristics is also analyzed theoretically.   The experimental results confirm its superiority over existing methods.

The paper addresses an important problem, and the method appears effective both in theory and practice.  The writing can still be improved, especially by showing more examples and presenting an abbreviated version of the pseudo-code in Appendix F.


**Note From Pc:**

if the above contains the word "oral" or "spotlight" please see: "oral" presentation means -> notable-top-5% and "spotlight" means -> notable-top-25%. As stated in our emails, we are disassociating presentation type from AC recommendations

**Summary Of Ac-Reviewer Meeting:**

During the meeting, the reviewers and the AC recognized the merits of the paper in both theory and practice.  We also noted the new results added during the rebuttal.  There were concerns about the writing, and the fact that the experimental results are not that outstanding.  However, overall, the consensus is that the merits outweigh the drawbacks, and it offers an interesting and effective method.